# Jointly efficient encoding and decoding in neural populations

**Simone Blanco Malerba**[1,2]*, **Aurora Micheli**[1¤], **Michael Woodford**[3], **Rava Azeredo da Silveira**[1,4,5]*

**1** Laboratoire de Physique de l'Ecole Normale Supérieure, ENS, Université PSL, CNRS, Sorbonne Université, Université de Paris, Paris, France, **2** Institute for Neural Information Processing, Center for Molecular Neurobiology, University Medical Center Hamburg-Eppendorf, Hamburg, Germany, **3** Department of Economics, Columbia University, New York, New York, United States of America, **4** Institute of Molecular and Clinical Ophthalmology Basel, Basel, Switzerland, **5** Faculty of Science, University of Basel, Basel, Switzerland

¤ Current address: Delft University of Technology, Delft, the Netherlands
* simone.bmalerba@gmail.com (SBM); rava@iob.ch (RAS)

**Data Availability Statement:** Code, simulation results, notebooks for analysis and data to reproduce figures are available on github (https://github.com/simonebmalerba/neural_VAE.git).

## Abstract

The efficient coding approach proposes that neural systems represent as much sensory information as biological constraints allow. It aims at formalizing encoding as a constrained optimal process. A different approach, that aims at formalizing decoding, proposes that neural systems instantiate a generative model of the sensory world. Here, we put forth a normative framework that characterizes neural systems as jointly optimizing encoding and decoding. It takes the form of a variational autoencoder: sensory stimuli are encoded in the noisy activity of neurons to be interpreted by a flexible decoder; encoding must allow for an accurate stimulus reconstruction from neural activity. Jointly, neural activity is required to represent the statistics of latent features which are mapped by the decoder into distributions over sensory stimuli; decoding correspondingly optimizes the accuracy of the generative model. This framework yields in a family of encoding-decoding models, which result in equally accurate generative models, indexed by a measure of the stimulus-induced deviation of neural activity from the marginal distribution over neural activity. Each member of this family predicts a specific relation between properties of the sensory neurons—such as the arrangement of the tuning curve means (preferred stimuli) and widths (degrees of selectivity) in the population—as a function of the statistics of the sensory world. Our approach thus generalizes the efficient coding approach. Notably, here, the form of the constraint on the optimization derives from the requirement of an accurate generative model, while it is arbitrary in efficient coding models. Moreover, solutions do not require the knowledge of the stimulus distribution, but are learned on the basis of data samples; the constraint further acts as regularizer, allowing the model to generalize beyond the training data. Finally, we characterize the family of models we obtain through alternate measures of performance, such as the error in stimulus reconstruction. We find that a range of models admits comparable performance; in particular, a population of sensory neurons with broad tuning curves as observed experimentally yields both low reconstruction stimulus error and an accurate generative model that generalizes robustly to unseen data.

**Funding:** This work was supported by the Alfred P. Sloan Foundation through grant G-2020-12680 to R. A. S. and M. W., the CNRS through grant UMR8023 to R. A. S., and the Simons Foundation Autism Research Initiative through grant 982347 to S. B. M. The funders had no role in study design, data collection and analysis, decision to publish, or preparation of the manuscript.

**Competing interests:** The authors have declared that no competing interests exist.

## Author summary

Our brain represents the sensory world in the activity of populations of neurons. Two theories have addressed the nature of these representations. The first theory—efficient coding—posits that neurons encode as much information as possible about sensory stimuli, subject to resource constraints such as limits on energy consumption. The second one—generative modeling—focuses on decoding, and is organized around the idea that neural activity plays the role of a latent variable from which sensory stimuli can be simulated. Our work subsumes the two approaches in a unifying framework based on the mathematics of variational autoencoders. Unlike in efficient coding, which assumes full knowledge of stimulus statistics, here representations are learned from examples, in a joint optimization of encoding and decoding. This new framework yields a range of optimal representations, corresponding to different models of neural selectivity and reconstruction performances, depending on the resource constraint. The form of the constraint is not arbitrary but derives from the optimization framework, and its strength tunes the ability of the model to generalize beyond the training example. Central to the approach, and to the nature of the representations it implies, is the interplay of encoding and decoding, itself central to brain processing.

## Introduction

Normative models in neuroscience describe stimulus representation and information transmission in the brain in terms of optimality principles. Among these, the efficient coding principle [1] posits that neural responses are set so as to maximize the information about external stimuli, subject to biological resource constraints. Despite this minimal assumption, this hypothesis has been successful in predicting neural responses to natural stimuli in various sensory areas [2–7]. The approach consists in specifying an *encoding* model as a stochastic map between stimuli and neural responses. The parameters of this model are then chosen so as to optimize a function that quantifies the coding performance, e.g., the mutual information between stimuli and neural responses. This optimization is carried out under a metabolic cost proportional, e.g., to the energy needed to emit a spike [8, 9]. The decoding process is assumed to be ideal and is carried out in a Bayesian framework: prior knowledge about the environment is combined with the evidence from neural activity to form a posterior belief about the stimulus [10–12].

The idea that the brain is capable of manipulating probabilities and uncertainty dates back to Helmoltz's view of perception as an inference process, in which the brain holds an internal statistical model of sensory inputs [13]. Mathematically, such an internal model can be formalized as a generative model in which stimuli are simulated by sampling from a distribution conditioned by one of a set of 'latent,' elementary features [14, 15]. In the generative process, a realization of latent features, i.e., a neural activity pattern, is associated to a distribution of stimuli, i.e., a likely sensory experience, in the form of a conditional probability distribution. These features can be chosen so as to allow for a semantic interpretation, such as oriented edges or textures in generative models of natural images [16–18], but this does not have to be the case [19]. It is then assumed that the role of sensory areas is to perform statistical inference by computing the posterior distribution over the latent features conditioned on the sensory observation, thereby 'inverting' the internal model. This posterior distribution is assumed to be represented in the neural activity, and different representation schemes have been proposed

[20–22]. Thus, as opposed to the efficient coding approach, which prescribes a stochastic mapping from stimulus to neural activity, the generative model approach prescribes the inverse process: from latent features, encoded in neural activity, to (simulated) sensory stimuli.

Here, we consider an extended efficient coding approach: while, typically, only the sensory encoding process is optimized, we consider jointly the encoding and decoding processes. In addition to a class of encoding transformations from stimuli to neural responses in a sensory area, we assume a class of generative models implemented downstream. These define maps from neural activity patterns, corresponding to realizations of the latent variables, to distributions over stimuli. Optimality is achieved when the generative distribution matches the true distribution of stimuli in the environment. If one assumes that the encoder and the decoder are jointly optimized in this framework, the system takes on the structure of a variational auto-encoder (VAE) [23–25].

Similarly to the classical efficient coding framework, here the encoder is set so as to maximize a variational approximation to the mutual information between stimuli and neural responses under a constraint on the neural resources. However, an important aspect of this formulation is that the constraint, rather than being imposed by hand, is a direct consequence of the assumption of an optimal internal model. This constraint is obtained as the statistical distance between the stimulus-evoked distribution of neural activity and the marginal distribution of neural activity assumed by the generative model. Furthermore, solutions do not require knowledge of the distribution of stimuli, as is the case in models of efficient coding, which posit optimal decoding through Bayesian inversion of the encoding process; here, solutions are learned on the basis of a limited volume of data samples [26, 27].

We apply our theoretical framework to the study of a population coding model with neurons with classical, bell-shaped tuning curves. By capitalizing on recent advances in the VAE literature, we solve the optimization problem as a function of the constraint on neural resources: we obtain a family of solutions which yield equally accurate generative models [28]. However, these solutions make different predictions on the corresponding neural representations, with different arrangements of tuning curves and statistics of prior over neural activity, as well as different predictions on the nature of coding and generalization. Related approaches have been explored in the literature, and predictions about the optimal allocation of coding resources, i.e., the tuning curves, as a function of the stimulus distribution have been derived, both at the single cell [2, 4, 29, 30] and at the population level [10, 31–35]. We examine how, in our framework, the optimal allocation of coding resources as a function of the statistics of stimuli varies as a function of the constraint. We show that our results subsume earlier predictions. In particular, we observe that solutions with broad tuning curves emerge without further constraints imposed by hand. These solutions feature coding performance which captures the statistics of the stimulus distribution and generalizes beyond the training dataset. Our results illustrate the way in which the interplay between encoder and decoder can shape the neural representations of sensory stimuli.

## Materials and methods

In what follows, we denote vectors in bold font and scalars in regular font. We denote by $\langle f(z) \rangle_{p(z)}$ the expectation of a function $f$ of a random variable $z$ distributed according to $p(z)$,

$$\langle f(z) \rangle_{p(z)} = \int dz\, p(z) f(z). \tag{1}$$

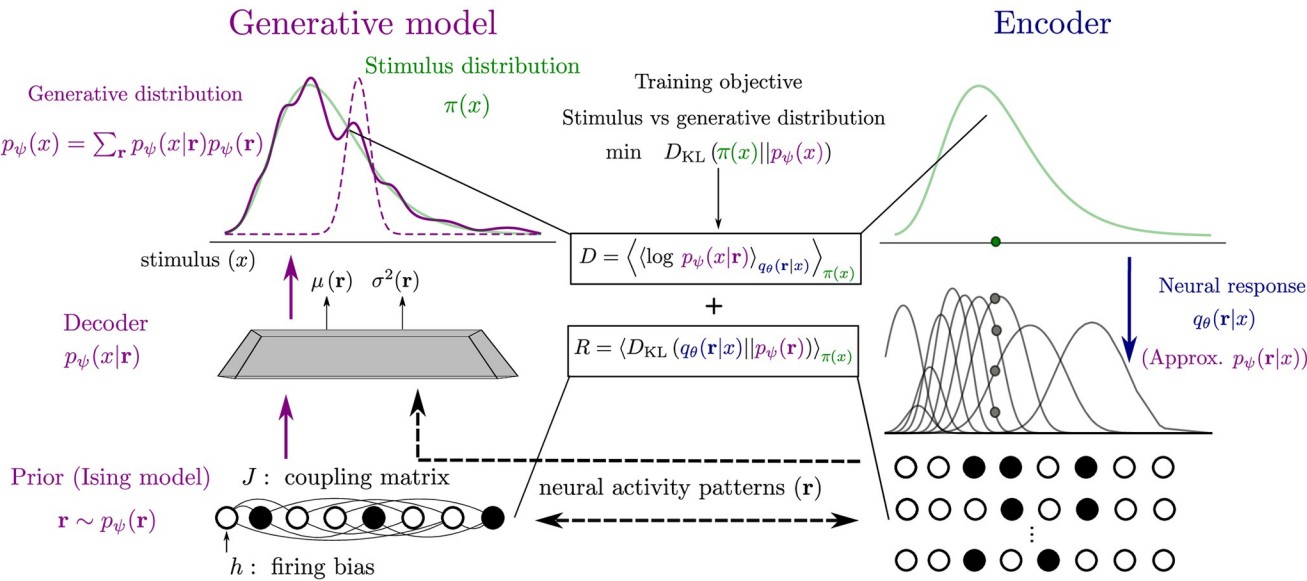

**Fig 1. Model architecture.** Left: generative model. In the generative model (purple), a neural activity pattern (white and black circles), sampled from the prior distribution (Ising model), $p_\psi(\mathbf{r})$, is mapped by the decoder to a probability distribution over stimuli, $p_\psi(x|\mathbf{r})$ (dashed purple curve), parametrized by functions of the activity patterns (here, mean and variance of a Gaussian distribution). The sum of the decoder output distributions, weighted by the prior over neural activity, defines the generative distribution (solid purple curve). The objective of the generative model is to produce stimuli according to the stimulus distribution in the environment, $\pi(x)$ (green curve). In order to do so, the posterior distribution over neural activity, given external stimuli, is approximated by an encoder, $q_\theta(\mathbf{r}|x)$ (right, blue). Neurons emit spikes according to bell-shaped tuning curves (grey curves) in response to a stimulus, $x$ (green dot), drawn from the stimulus distribution $\pi(x)$ (green curve). The population response consists in a binary neural activity pattern, $\mathbf{r}$ (white and black circles). The two models are trained such as to match the generative and stimulus distributions; this objective is approximated by minimizing two loss functions. The distortion, $D$, maximizes the likelihood of the observed stimulus after the encoding-decoding process (upper dashed black arrow). The rate, $R$, minimizes the Kullback-Leibler divergence between the conditional encoding distribution and the marginal (prior) distribution of neural activity of the generative model (bottom dashed black arrow).

## Generative model (decoder)

We assume that the brain holds an internal generative model of the environment (Fig 1, left). This model is specified by the joint probability of the neural activity of $N$ neurons, $\mathbf{r} = \{r_1, \ldots, r_N\}$, and a scalar sensory stimulus, $x$, $p_\psi(\mathbf{r}, x)$, where $\psi$ denotes a set of parameters. The neural activity is viewed as a latent variable, sampled from a prior distribution, $p_\psi(\mathbf{r})$. Given the neural activity, or latent state, a 'decoder' maps it to a distribution over stimuli, $p_\psi(x|\mathbf{r})$. This process specifies likely sensory experiences corresponding to a neural activity pattern, according to the generative model. We assume that the same generative process, which outputs a simulated quantity, $x$, is used by the brain when, e.g., producing an estimate in the context of a behavioral task.

We consider neural activity over short time intervals, such that each neuron either emits one spike or is silent; this assumption is valid in scenarios when sensory coding occurs on a short time and have been largely studied in the literature (see, e.g., work on maximum entropy models [36, 37] or generalized linear models [38].) The set of possible activity patterns is then the set of binary vectors, $\mathbf{r} = (r_1, r_2, \ldots, r_N)$, where $r_i \in \{0, 1\}$; in what follows, the sum $\sum_\mathbf{r} \cdot$ denotes the sum over these $2^N$ binary patterns. We model the prior distribution over neural activity as the maximum-entropy distribution constrained by the first- and second-order statistics, a model which has been proposed as a model of the distribution of activity in neural systems, e.g., in retina and in cortex [36]. In the case of binary patterns, this maximum-entropy

distribution takes the form of an Ising model, or Boltzmann machine,

$$p_\psi(\mathbf{r}) = \exp\left(\mathbf{h}^T\mathbf{r} + \mathbf{r}^T J\mathbf{r} - \log Z\right), \tag{2}$$

where $\mathbf{h}$ is the vector of biases, $J$ is the matrix of couplings (with our choice of parametrization, the diagonal elements of $J$ vanish), and $Z = \sum_\mathbf{r} \exp(\mathbf{h}^T\mathbf{r} + \mathbf{r}^T J\,\mathbf{r})$ is a normalization constant (also called partition function). Throughout the paper, we employ Eq (2) as the prior distribution of neural activity, which allows for correlations among neurons. The only exception is in S3 Fig, where we illustrate the consequences of choosing a less flexible distribution, i.e., a product of $N$ independent Bernoulli distributions, obtained by setting to 0 all entries of the coupling matrix, $J$.

On the basis of experimental findings, it has been suggested that the brain encodes a probability distribution of the stimulus, rather than a simple point estimate [20, 39]. We follow this view by assuming that the generative model encodes a probability distribution over stimuli in a parametric form, with parameters obtained as flexible functions of the neural activity pattern. We consider two forms of generative distributions. First, we model the generative distribution as a simple Gaussian, by assuming that the decoder maps activity patterns to a mean and a variance,

$$p_\psi(x|\mathbf{r}) = \mathcal{N}(\mu_\phi(\mathbf{r}), \sigma_\phi(\mathbf{r})); \tag{3}$$

we parameterize these functions as two-layer neural networks, and we denote by $\phi$ the set of weights and biases in the network. Second, to account for heavy-tail behavior of some natural stimulus distributions, we posit that the brain represents mean and variance in a logarithmic scale. Thus, we also consider a log-normal distribution whose mean and variance of the logarithm of the stimulus values are parametrized as two-layer neural networks,

$$p_\psi(x|\mathbf{r}) = \mathcal{LN}(\mu_\phi(\mathbf{r}), \sigma_\phi(\mathbf{r})). \tag{4}$$

The parameters of the generative distribution and of the prior, $\psi = \{\phi, \mathbf{h}, J\}$, constitute the full set of parameters of the generative model.

With this parametrization choices, for $N$ neurons the distribution $p_\psi(x) = \sum_\mathbf{r} p_\psi(x|\mathbf{r})p_\psi(\mathbf{r})$ is a mixture of $2^N$ normal or log-normal distributions, and is thus highly flexible. (In the Gaussian case, it has been shown that a well-chosen Gaussian mixture can be used to approximate any smooth density function [40, 41].) We will denote the decoder as 'Gaussian' or 'log-normal,' depending on whether we use Eqs (3) or (4) as the decoding distribution, respectively. We note that the generative distribution is a member of the exponential family, whose natural parameters are non-linear functions of the latent variables. This represents a generalization of the classic Helmoltz machine, that takes into account higher-order sufficient statistics [14, 19].

## Recognition model (encoder)

The generative model assumes a prior distribution over neural activity, $p_\psi(\mathbf{r})$: typically neural activity changes as a function of external stimuli to encode information about the sensory world. The recognition model prescribes the way in which stimuli evoke (are mapped to) neural activity, through a conditional distribution $q_\theta(\mathbf{r}|x)$. In the next section, we relate these two models by optimizing a loss function which takes both account.

We consider a population of $N$ neurons which responds to a continuous scalar sensory stimulus, $x$, distributed according to a prior distribution, $\pi(x)$ (Fig 1, right). In order to avoid confusion with the prior distribution over neural activity patterns, $p_\psi(\mathbf{r})$, defined above, we will refer to $\pi(x)$ as the data, or stimulus, distribution. The encoding distribution is the conditional probability distribution over neural activity patterns given the stimulus, $q_\theta(\mathbf{r}|x)$, where $\theta$

denotes the set of parameters. For the sake of simplicity, we assume that neurons spike independently conditional on the stimulus, such that $q_\theta(\mathbf{r}|x) = \prod_{i=1}^N q_\theta(r_i|x)$. (We focus on conditionally independent neural activity because modeling correlations would require additional $\mathcal{O}(N^2)$ parameters and further assumptions about their structure (e.g., related to the connectivity that gives rise to the neural activity). As it has been extensively documented in the literature, the same coding performance can be achieved with different configurations of signal and noise correlations [42, 43]. Allowing for noise correlations would complicate the interpretation of our results.)

We consider a Bernoulli distribution, with the probability of spiking of a neuron obtained as

$$q_\theta(r_i = 1|x) = \frac{f_i(x)}{1 + f_i(x)}, \tag{5}$$

where $f_i(x)$ is referred to as the neuron's tuning curve. (This model can also be viewed as the limit of a Poisson model for spiking neurons with low rates [44, 45].) We parametrize tuning curves as Gaussian functions, a shape widely observed in early sensory areas, as

$$f_i(x) = A_i \exp\left(-\frac{(x - c_i)^2}{2w_i^2}\right), \tag{6}$$

with $c_i$ the preferred stimulus of neuron $i$, $w_i$ the tuning width, and $A_i$ the amplitude. Thus, the probability of spiking of a neuron can be written as $q_\theta(r_i = 1|x) = \mathcal{S}(\eta_i(x))$, with $\eta_i(x) = \frac{-(x - c_i)^2}{2w_i^2} + \log A_i$ and $\mathcal{S}(y) = 1/(1 + \exp(-y))$, the logistic function. (We note the analogy between this encoder and the classic recognition model of the Helmoltz machine, with the difference that here we employ a quadratic function of the stimulus as the argument of the sigmoid, while a linear function is employed in the classic Helmoltz machine.) In the canonical form of the exponential family, the resulting multivariate Bernoulli distribution can be written as

$$q_\theta(\mathbf{r}|x) = \exp\left(\boldsymbol{\eta}(x)^T \mathbf{r} - \sum_{i=1}^N \log\left(1 + e^{\eta_i(x)}\right)\right), \tag{7}$$

with $\boldsymbol{\eta}(x) = (\eta_1(x), \ldots, \eta_N(x))$ the vector of natural parameters and $\theta = \{A_i, c_i, w_i\}_{i=1}^N$ the set of parameters of the encoder.

## Training objective

We introduced two models of joint distributions of neural activity and sensory stimulus, the generative model, $p_\psi(x|\mathbf{r})p_\psi(\mathbf{r})$, and the recognition model, $q_\theta(\mathbf{r}|x)\pi(x)$. Optimality requires that their joint distributions be matched. To approach this condition, we set the parameters of the generative and recognition model so as to minimize the Kullback-Leibler divergence between the two joint distributions,

$$\min_{\psi,\theta} D_{\mathrm{KL}}(q_\theta(\mathbf{r}|x)\pi(x)||p_\psi(x|\mathbf{r})p_\psi(\mathbf{r})). \tag{8}$$

This quantity, is an upper bound to the Kullback-Leibler divergence between the true stimulus distribution and the marginal generative distribution over stimuli: we have

$$D_{\mathrm{KL}}(q_\theta(\mathbf{r}|x)\pi(x)||p_\psi(x|\mathbf{r})p_\psi(\mathbf{r})) = D_{\mathrm{KL}}(\pi(x)||p_\psi(x)) + \langle D_{\mathrm{KL}}(q_\theta(\mathbf{r}|x)||p_\psi(\mathbf{r}))\rangle_{\pi(x)}, \tag{9}$$

where the second term on the right-hand side is non-negative by definition of the Kullback-Leibler divergence. Eq (9) implies that, by optimizing Eq (8), we minimize an upper bound on the divergence between the true distribution of stimuli and the generative distribution (first term on the right-hand side, purple and green curves in the top-left panel in Fig 1). Furthermore, this quantity, up to a constant term, is equal to the negative log-likelihood of the observed data under the generative model, $\langle \log p_\psi(x) \rangle_{\pi(x)}$, a quantity called 'evidence' and often used as a maximization objective in machine learning frameworks [23, 24] (see Discussion below and Eq (14)).

The quantity in Eq (8) can be rewritten as

$$D_{\text{KL}}(q_\theta(\mathbf{r}|x)\pi(x)||p_\psi(x|\mathbf{r})p_\psi(\mathbf{r})) = H(\pi) - \left\langle \sum_{\mathbf{r}} q_\theta(\mathbf{r}|x) \log p_\psi(x|\mathbf{r}) \right. \tag{10}$$
$$\left. + D_{\text{KL}}(q_\theta(\mathbf{r}|x)||p_\psi(\mathbf{r})) \right\rangle_{\pi(x)},$$

where we recognize in the last two terms of the right-hand side the opposite of the so-called 'evidence lower bound' (ELBO). Thus, by ignoring the stimulus entropy that does not depend on the parameters, the problem in Eq (8) can be formulated as

$$\min_{\{\psi,\theta\}} \{-\text{ELBO} = D + R\}. \tag{11}$$

Borrowing the nomenclature from rate-distortion theory, we call *distortion* the quantity

$$D = \left\langle -\sum_{\mathbf{r}} q_\theta(\mathbf{r}|x) \log p_\psi(x|\mathbf{r}) \right\rangle_{\pi(x)}, \tag{12}$$

equal to the second term on the right-hand-side of Eq (10), and *rate* the quantity

$$R = \langle D_{\text{KL}}(q_\theta(\mathbf{r}|x)||p_\psi(\mathbf{r})) \rangle_{\pi(x)} = \left\langle \sum_{\mathbf{r}} q_\theta(\mathbf{r}|x) \log \left( \frac{q_\theta(\mathbf{r}|x)}{p_\psi(\mathbf{r})} \right) \right\rangle_{\pi(x)}, \tag{13}$$

equal to the third term. This framework goes by the name of variational autoencoder (VAE) [23, 24]. The encoder maps a stimulus sample, $x$, to a neural activity pattern, $\mathbf{r}$, according to $q_\theta(\mathbf{r}|x)$. The activity pattern corresponds to a realization of the latent variable in the generative model, and is mapped back ('decoded') to a distribution over stimuli according to $p_\psi(x|\mathbf{r})$. As one typically does not have access to the true data distribution, but only to a set of samples, the average over $\pi(x)$ is approximated by an empirical average over a set of $N_{trn}$ samples, $\langle f(x) \rangle_{\pi(x)} \approx \sum_{i=1}^{N_{trn}} f(x_i)/N_{trn}$.

In machine learning, the derivation of the VAE is often presented from a different starting point. The generative model parameters are required to maximize the evidence, i.e., the log-likelihood of data samples under the generative process, $\langle \log p_\psi(x) \rangle_{\pi(x)}$, which is equivalent, up to a constant term corresponding the data entropy, to the first term on the right-hand side of Eq (9),

$$\langle \log p_\psi(x) \rangle_{\pi(x)} = -H(\pi) - D_{\text{KL}}(\pi(x)||p_\psi(x)). \tag{14}$$

This, however, requires the inversion of the generative model, to obtain $p_\psi(\mathbf{r}|x)$, a typically intractable task. A common approach is to introduce a variational approximation of this posterior distribution, $q_\theta(\mathbf{r}|x)$, and, by combining the identities in Eqs (14), (10) and (9), to maximize the quantity $-(D + R)$ which constitutes a (variational) lower bound to the evidence. Yet an alternative derivation is obtained by applying Jensen's inequality to the evidence rewritten

in such a way to feature the variational approximation, $\langle \log p_\psi(x) \rangle_{\pi(x)} = \langle \log \int d\mathbf{r} q_\theta(\mathbf{r}|x) p_\psi(x, \mathbf{r}) / q_\theta(\mathbf{r}|x) \rangle_{\pi(x)}$ [23–25].

## Mean-squared error

We note that, due to the fact that the variance of the generative distribution depends on the neural responses, the distortion differs from the more usual mean-squared error (MSE) loss function of classical autoencoders, also commonly employed to measure the performance of neural codes. Indeed, in the case of a Gaussian decoder, the distortion function is written as

$$D = \left\langle \sum_{\mathbf{r}} q_\theta(\mathbf{r}|x) \left( \frac{(\mu_\phi(\mathbf{r}) - x)^2}{2\sigma_\phi^2(\mathbf{r})} + \frac{1}{2} \exp\left(2\pi\sigma_\phi^2(\mathbf{r})\right) \right) \right\rangle_{\pi(x)}, \tag{15}$$

while the MSE, if we assume that the brain has access to the full (approximation) of the posterior distribution over stimuli, $p_\psi(x|\mathbf{r})$, is obtained as

$$\varepsilon^2 = \left\langle \sum_{\mathbf{r}} q_\theta(\mathbf{r}|x)(\mu_\phi(\mathbf{r}) - x)^2 \right\rangle_{\pi(x)}, \tag{16}$$

where we have used the maximum a posteriori (MAP) estimate, $\hat{x}_{MAP} = \mu_\phi(\mathbf{r})$. In the case of a log-normal decoder, the distortion is obtained as

$$D = \left\langle \sum_{\mathbf{r}} p(\mathbf{r}|x) \left( \frac{(\mu_\phi(\mathbf{r}) - \log x)^2}{2\sigma_\phi^2(\mathbf{r})} + \frac{1}{2} \log\left(2\pi x \sigma_\phi^2(\mathbf{r})\right) \right) \right\rangle_{\pi(x)}, \tag{17}$$

where the MAP estimate is given by $\hat{x}_{MAP} = \exp\left(\mu_\phi(\mathbf{r}) - 2\sigma_\phi^2(\mathbf{r})\right)$. (Note that we quote instead the square root of the MSE (RMSE) in comparing the predictions of our model to the data on acoustic frequency-difference limens, in the section "Case study: neural encoding of acoustic frequencies.").

If we assume that the brain makes use of the same generative model to produce behavioral responses (i.e., does not have access to an auxiliary (ideal) decoder), a straightforward operation is to draw samples, rather than computing statistics such as the mode. Thus, in the Results section, we also consider the MSE obtained when the stimulus estimate, $\hat{x}$, is sampled from the posterior distribution, $\hat{x} \sim p_\psi(x|\mathbf{r})$, as

$$\begin{aligned} \varepsilon_{\text{sampling}}^2 &= \left\langle \sum_{\mathbf{r}} q_\theta(\mathbf{r}|x) \int d\hat{x} p_\psi(\hat{x}|\mathbf{r})(\hat{x} - x)^2 \right\rangle_{\pi(x)} \\ &= \left\langle \sum_{\mathbf{r}} q_\theta(\mathbf{r}|x) \left[ (\mu_\phi(\mathbf{r}) - x)^2 + \sigma_\phi^2(\mathbf{r}) \right] \right\rangle_{\pi(x)}. \end{aligned} \tag{18}$$

We note that since our decoder is non-ideal and might be biased, the Fisher information of the encoder distribution, often used to quantify the encoding properties of neural populations as it provides a lower bound to the variance of any unbiased estimator [31], here yields a poor prediction of the coding performance. In S6 Fig, we compare the MSE with the inverse of the Fisher information, which in the case of independent Bernoulli neurons can be obtained as

[10]

$$J(x) = \sum_{j=1}^{N} \frac{(x - c_j)^2}{\sigma_j^4} \frac{1}{\left(1 + \exp(-\eta_j(x))\right)^2},$$

(19)

with $\eta_j(x)$ defined as in Eq (7).

## Constrained optimization and connection with efficient coding

It is a known issue in the VAE literature that, when the generative distribution is flexible given the data distribution (meaning that $p_\psi(x|\mathbf{r})$ has enough degrees of freedom to approximate $\pi(x)$), the ELBO optimization problem exhibits multiple solutions. Optimization algorithms based on stochastic gradient descent are biased towards solutions with low rate and high distortion, a phenomenon which goes by the name of posterior collapse [28, 46]. In the extreme case, the model relies entirely on the power of the decoder and ignores the latent variables altogether: all realizations of the latent variables are mapped to the data distribution, $p_\psi(x|\mathbf{r}) \approx \pi(x)$, and, consequently, all stimuli are mapped to the same representation, $q_\theta(\mathbf{r}|x) \approx p_\psi(\mathbf{r})$.

We overcome this issue by addressing a related, constrained optimization problem. We minimize the distortion subject to a maximum, or 'target,' value of the rate, $\bar{R}$:

$$\min_{\{\theta,\psi\}} \quad D$$
$$\text{subject to} \quad R \le \bar{R}.$$

(20)

The set of parameters $\{\theta, \psi\}$ that satisfy the constraint $R \le \bar{R}$ is called feasible set. By writing the associated Lagrangian function with multiplier $\beta \ge 0$, we have that

$$\max_{\beta \ge 0} \{L(\theta, \psi, \beta) = D + \beta(R - \bar{R})\} = \begin{cases} D & \text{if } \{\theta, \psi\} \text{ is feasible} \\ \infty & \text{otherwise} \end{cases}.$$

(21)

Solutions of Eq (20) can thus be found as solutions to the 'minimax' problem,

$$\min_{\{\theta,\psi\}} \max_{\beta \ge 0} \{L(\theta, \psi, \beta) = D + \beta(R - \bar{R})\}.$$

(22)

The Lagrangian has a form similar to that of the negative ELBO, with an additional $\beta$ factor multiplying the rate; this framework was presented as an extension of the classical VAE, with the aim of obtaining disentangled latent representations, in Refs. [28, 47].

Before addressing the optimization problem, we note that the two terms contributing to the ELBO are related to the mutual information of stimuli and neural responses of the encoder,

$$I_{enc}(\mathbf{r}, x) = \left\langle \log \frac{q_\theta(\mathbf{r}, x)}{\pi(x) q_\theta(\mathbf{r})} \right\rangle_{q_\theta(\mathbf{r}, x)},$$

(23)

through the bounds

$$H(\pi) - D \le I_{enc}(\mathbf{r}, x) \le R,$$

(24)

where $H(\pi)$ is the entropy of the stimulus distribution [28]. The two inequalities arise because in the variational approximation the posterior over stimuli, $p_\psi(x|\mathbf{r})$, replaces $q_\theta(x|\mathbf{r})$, and the prior over activity patterns, $p_\psi(\mathbf{r})$, replaces $q_\theta(\mathbf{r})$, respectively. Since we are considering continuous stimuli, $H$ is a differential entropy, and is thus defined up to a constant, and $D$ can take negative values. Below, we will illustrate the properties of the generative model also through

the $D_{\mathrm{KL}}$ divergence between the stimulus and the marginal generative distribution, i.e., the first term on the right-hand side of Eq (9), which is non-negative.

Eq (24) has two important consequences. First, it allows us to interpret the problem in Eq (20) as an efficient coding problem, where the objective is to maximize a lower bound to the mutual information, $H - D$, subject to a bound on the neural resources, $\bar{R}$. Contrary to the classical efficient coding literature, in which a metabolic constraint is imposed by hand, here it results from the original formulation of the problem as optimization of the ELBO, and it is affected by the assumptions made on the generative model (more specifically, on the prior distribution). The rate is minimized when the stimulus has little impact on the distribution of neural responses. A metabolic constraint is similarly bound to disfavor large changes in neural activity induced by stimuli. For this reason, the rate can interpreted as an abstract form of metabolic constraint.

Second, it prescribes limiting values to the solutions of Eq (20). If the variational distributions, $p_\psi(\mathbf{r})$ and $p_\psi(x|\mathbf{r})$, are flexible enough to approximate $q_\theta(\mathbf{r})$ and $q_\theta(x|\mathbf{r})$, we can achieve both inequalities, and we have $D = H - \bar{R}$. Along this line in the rate-distortion plane, the negative ELBO achieves its minimum value, equal to the stimulus entropy. In what follows, we call 'optimal' the solutions of Eq (20) and 'lowest limiting values' the values on the line of ideal solutions, $D = H - \bar{R}$.

We address the minimax problem of Eq (22) numerically through a two-timescales, alternated stochastic gradient descent-ascent, Alg. 1. We denote by $\{\theta^*, \psi^*, \beta^*\}$ the optimal parameters. It is possible to show under some assumptions that (*i*) the parameter set $\{\theta^*, \psi^*\}$ is a solution of the problem defined in Eq (20); (*ii*) if $\beta^* > 0$, the constraint on the rate is satisfied as an equality, $R = \bar{R}$, and $\beta^*$ is the negative slope of the curve of minimum distortion as a function of the target rate, $\frac{dD}{dR}\big|_{\theta^*,\psi^*} = -\beta^*$; (*iii*) if $\beta^* = 1$, then the parameters $\{\theta^*, \psi^*\}$ maximize the ELBO. We report the conditions under which these statements are true, and their proofs, in the section S1 Appendix.

**Algorithm 1** Two-timescale optimization algorithm.
```
1: Inputs: target rate R̄, dataset 𝒟
2: Initialize: β = 1, encoder/decoder parameters= {θᵢ, ψᵢ}
3: while convergence do
4:   Define β-ELBO: L_β = D + βR
5:   for batch in 𝒟 do
6:     Update parameters: (θ, ψ) ← Adam (∇_θL_β(batch), ∇_ψL_β(batch))
7:   end for
8:   β → max{β + η_β(R - R̄), 0}
9: end while
10: return
```

## Numerical optimization and related computations

Numerical simulations are carried out using PyTorch. We solve the optimization problem in Eq (22) through stochastic gradient descent on the loss on a training dataset with $N_{trn}$ samples from $\pi(x)$. Except for the results shown in the section "Generalization and role of the rate as regularizer", we used $N_{trn} = 2000$, divided in minibatches of size 128, with the Adam optimizer [48] with learning rate equal to $10^{-4}$ and otherwise standard hyperparameters. The learning rate for $\beta$, $\eta_\beta$, is set to 0.1. The training is iterated over multiple passes over the data (epochs) with a maximum of 5000 epochs and it is stopped when the training loss running average remains unchanged (with a tolerance of $10^{-5}$) for 100 consecutive epochs. The properties of the model are then quantified and illustrated using another dataset of $N_{tst} = 2000$ samples from $\pi(x)$. Except for the results shown in in the section "Generalization and role of the rate as

regularizer", due to the volume of the training set, the metrics (e.g., distortion, rate, MSE) recorded using the training and the test set are indistinguishable, and we refer to them without making the distinction explicit. In the section "Generalization and role of the rate as regularizer" we explore the issue of generalization, and we use different values of $N_{trn}$; we adapt the size of the minibatch to keep it approximately in proportion to the training set size. Here, the metrics recorded on the test set deviate from those obtained from the training set, and we differentiate them by using the 'trn' and 'tst' suffixes.

The parameters are initialized as follows. The preferred positions, $c_i$, are initialized as the centroids obtained by applying a $k$-means clustering algorithm (with $k = N$) to the set of stimuli in the dataset. Tuning widths are initialized by setting $w_i = |c_i - c_j|$, with $c_j$ the closest preferred position to $c_i$, and the amplitude is set equal to 1, corresponding to a maximum probability of spiking of 0.5. Random noise of small variance is applied to the initial value of the parameters. The biases of the prior, $\mathbf{h}$, are set equal to 1. The coupling matrix, $J$, is sampled from a Wishart distribution with $n = p = N$ degrees of freedom, with variance equal to $1/N$ and diagonal elements set to 0. The weights of the decoder neural network are initialized with the standard Kaiming initialization, corresponding to a uniform distribution between $a$ and $-a$, with $a = \sqrt{6/n_{l-1}}$, and $n_{l-1}$ the size of the input layer; the biases are set to 0. The results are averaged over 16 network initializations. An example of the evolution of $D$, $R$, and $\beta$ during training is illustrated in Fig 2.

We illustrate results for $N$ small enough so that it be possible to compute explicitly the sums over activity patterns appearing in the loss function. This also allows us to explore regimes in which the information is compressed in the activity of a finite population of neurons. In S2 Appendix, we discuss the numerical issues encountered when the population size is large, and we mention possible solutions.

For the results shown in the sections "Optimal allocation of neural resources and coding performance" and "Case study: neural encoding of acoustic frequencies", we use the function `curve_fit` from the `scipy` package to fit the MSE, the neural density, and the tuning width as functions of the stimulus probability with power laws. Concretely, we log-transform the data, and we fit a linear model, $\log f(x) = A - \gamma \log \pi(x)$, with $A$ and $\gamma$ as parameters. The MSE as a function of $x$ is obtained by averaging across 16 initializations of the model parameters. We computed the neural density by applying a Kernel density estimate to the centers of the tuning curves. In order to match the scale of the predicted error, $\varepsilon(f)$, to the scale of the

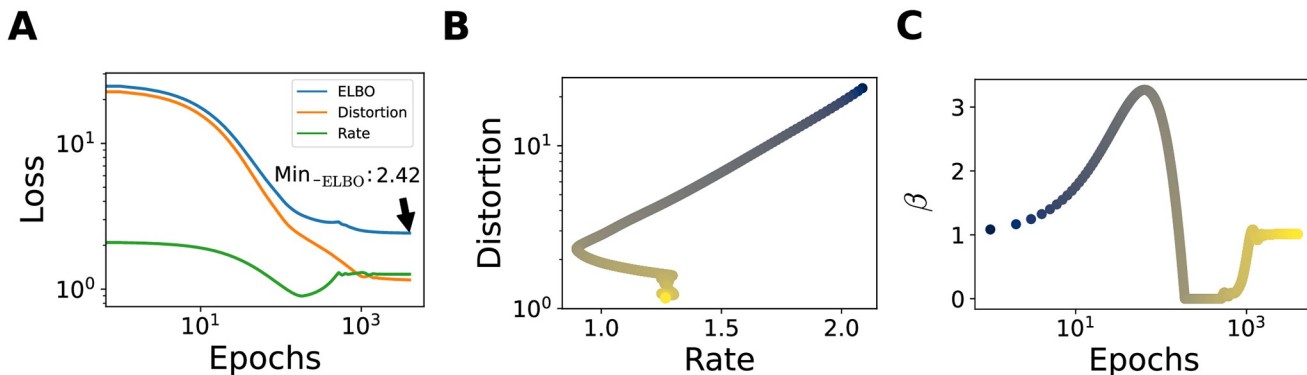

**Fig 2. Example of training.** $\pi(x) = \mathcal{LN}(1, 1)$, $N = 12$, $\bar{R} = 1.32$ (**A**) Evolution of negative ELBO, and the two terms, $D$ and $R$, with training epochs. Plot in log-log scale. (**B**) Joint evolution of $R$ and $D$ in the rate-distortion plane, colored according to the epoch (increasing from blue to yellow, colors in logarithmic scale). (**C**) Evolution of $\beta$ during training.

experimental data on frequency-difference limens, $y(f)$, we found the parameter $a$ which minimized the squared distance between the logarithm of the data and the logarithm of the mean error, across all data points, $\hat{a} = \arg\min_a \sum_f (\log y(f) - \log(\varepsilon(f)/a))^2$.

## Results

We optimize jointly an encoder, a population of neurons with simple tuning curves, which stochastically maps stimuli to neural activity patterns, and a decoder, a neural network which maps activity patterns, interpreted as latent variables, to distributions over stimuli. The system is set so as to minimize a bound to the Kullback-Leibler ($D_{KL}$) divergence between the generative distribution and the true distribution of stimuli (Fig 1). By formulating the training objective as a constrained optimization problem, we characterize the space of optimal solutions as a function of the value of the constraint; we then discuss the properties of the encoder and of the decoder in the family of solutions. This constrained optimization problem takes a form similar to that of the efficient coding problem:

$$\min_{\{\theta,\psi\}} \quad D$$
$$\text{subject to} \quad R \leq \bar{R}. \tag{25}$$

Here the minimization of the distortion, $D$, is the analog to the maximisation of the mutual information between stimuli and neural responses in efficient coding. The constraint on the rate, $R$, corresponds to a metabolic constraint in efficient coding, and quantifies the impact of a stimulus on the neural response (see Materials and methods). We mainly focus on the solutions of the asymptotic problem with large training set. In the section "Generalization and the role of the rate as regularizer," we address the implications of a finite training set.

### Degeneracy of optimal solutions

We begin by illustrating two alternative solutions of the ELBO optimization problem, Eq (11), characterized by different contributions of the two terms, $D$ and $R$. We first consider the simple, but instructive, case of a Gaussian distribution over stimuli, $\pi(x) = \mathcal{N}(\mu_p, \sigma_p^2)$, together with a Gaussian decoder. In order to minimize the rate, a possible solution is to set the parameters of the encoder so as to map all stimuli to the same distribution over neural activity patterns which in turn approximates the prior distribution, $q_\theta(\mathbf{r}|x) \approx p_\psi(\mathbf{r})$. This is achieved by neurons with low selectivity, i.e., with broad and overlapping tuning curves (Fig 3A, top). Despite the uninformative neural representation, a perfectly accurate generative model is obtained (in this special, Gaussian case) by mapping all activity patterns to the parameters of the data distribution, $\mu_\psi(\mathbf{r}) = \mu_p$ and $\sigma_\psi^2(\mathbf{r}) = \sigma_p^2$ for all $\mathbf{r}$; in this way, the generative distribution becomes independent from the neural activity, $p_\psi(x|\mathbf{r}) \approx \pi(x)$ (Fig 3B, top). The rate term becomes negligible and the distortion equal to the stimulus entropy, thereby satisfying the leftward inequality in Eq (24). The sum of the two terms is equal to the stimulus entropy, the lower bound of the negative ELBO; the neural representation, however, retains no information about the stimulus.

At the opposite extreme, it is possible to minimize the distortion by learning an injective encoding map that associates distinct stimuli to distinct activity patterns. In our framework, this is achieved by narrow and non-overlapping tuning curves that tile the stimulus space (Fig 3A, bottom). For a given encoding distribution, the optimal prior distribution which

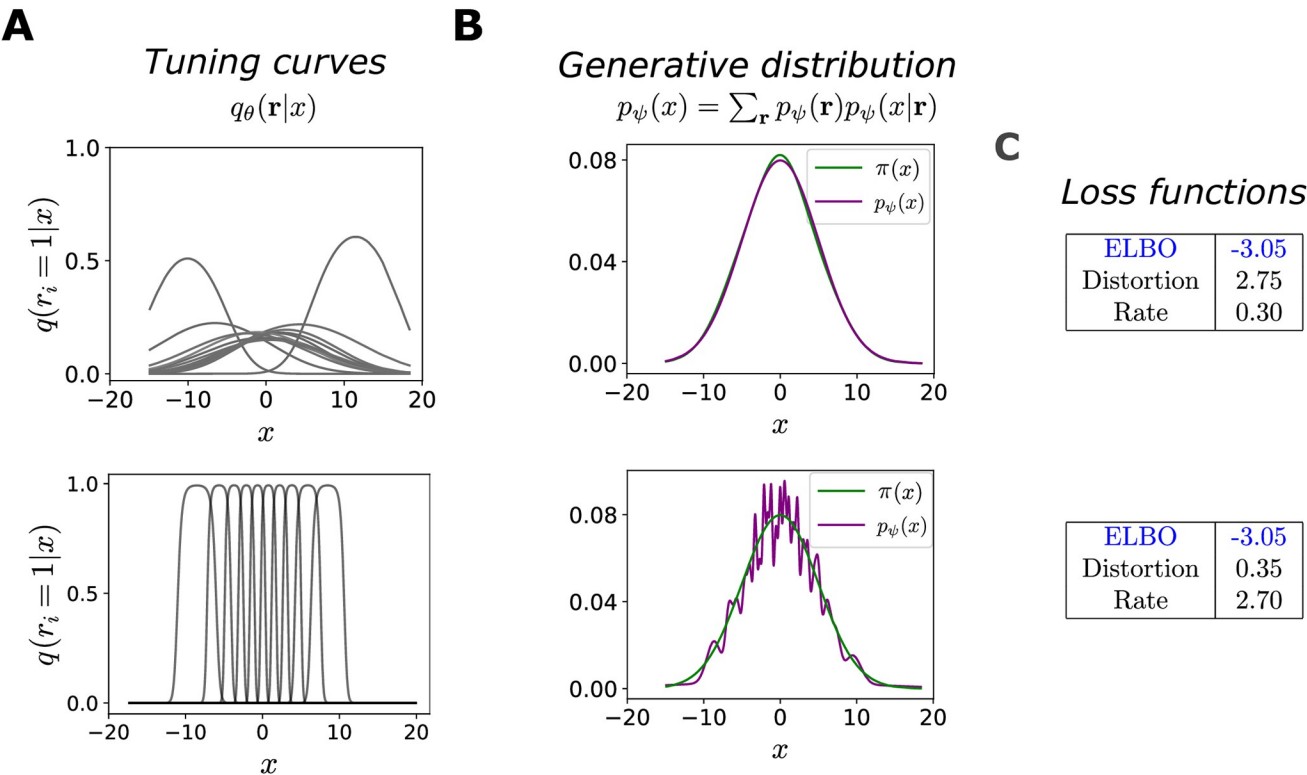

**Fig 3. Qualitatively different optimal configurations.** In all simulations, $N = 10$ and $\pi(x) = \mathcal{N}(0, 5)$. Top row: high-distortion, low-rate solution. Bottom row: low-distortion, high-rate solution. (**A**) Bell-shaped tuning curves of the encoder (probability of neuron $i$ to emit a spike, as a function of $x$). (**B**) Comparison between the stimulus distribution, $\pi(x)$ (green curve), and the generative distribution, $p_\psi(x) = \Sigma_\mathbf{r} \, p_\psi(x|\mathbf{r}) p_\psi(\mathbf{r})$ (purple curve). (**C**) Numerical values of the ELBO, and the distortion and rate terms.

minimizes the rate, Eq (13), is equal to the marginal encoding distribution,

$$q_{\psi^*}(\mathbf{r}) = \langle q_\theta(\mathbf{r}|x) \rangle_{\pi(x)}. \tag{26}$$

(See Ref. [49] for an application of this optimal prior in the context of VAEs.) If the encoding distribution is different for each stimulus, the rate term does not vanish, but, numerically, we find that the parameters of the prior can still be set so as to approximate Eq (26), achieving the rightward inequality in Eq (24). Furthermore, the decoder then maps each activity pattern to a narrow Gaussian distribution over stimuli, so as to suppress the distortion to negligible values. As a consequence, the negative ELBO again achieves its lower bound, and it is possible to obtain a generative model that approximates closely the stimulus distribution, though less smoothly (Fig 3B, bottom).

Although these two solutions yield comparable values of the ELBO (Fig 3C) and equally accurate generative models, the corresponding neural representations are utterly different. This case is special and contrived, because the conditional generative distribution has the same functional form as the stimulus distribution, and thus a perfect generative model is obtained even when it ignores the latent variables. However, the reasoning extends to more complex cases, and the choice of the forms of the decoding distribution and the prior determines the ability of the system to optimize the ELBO in different ways [28]. In order to achieve a small distortion at low rates, the generative distribution must be flexible enough to approximate the data distribution even when the latent variables carry little information about the stimulus.

Conversely, prior distributions which can fit marginal encoding distributions in which each data point is mapped precisely to a realization of the latent variables, achieve a low value of the rate for small distortion. Indeed, as we show next, we observe the existence of multiple solutions of the ELBO optimization problem for more complex stimulus distributions.

## Analysis of the family of optimal solutions

We explore systematically the space of solutions which optimize the ELBO, by minimizing the distortion subject to a constraint on the maximum ('target') value of the rate, $\bar{R}$, a formulation which yields a generalized objective function (Eq (22)) with a factor $\beta$ that weighs the rate term (see Materials and methods). The value of $\bar{R}$ is an upper bound to the mutual information between stimulus and neural response; it thereby imposes a degree of 'compression' of the information in the encoding process. We illustrate results for the simple, yet non-trivial, choice of a log-normal stimulus distribution and Gaussian decoder, which exhibits a similar degeneracy as the simple case described above, in spite of the discrepancy between the stimulus and generative distributions (S1 Fig). Furthermore, the degree of degeneracy of solutions is comparable to the case in which a log-normal decoder is used instead of Gaussian one (S2 Fig). Similar observations hold for other distributions as well: in S4 Fig we illustrate the case of a multimodal distribution.

Each solution is associated with a point $(\bar{R}, D)$ in the rate-distortion plane. By varying the value of $\bar{R}$, we trace the curve of the optimal distortion as a function of the target rate (Fig 4A). We focus on the range of values of $\bar{R}$ resulting in $\beta^* = 1$, for which $R = \bar{R}$ and the corresponding solutions also yield an optimal value of the ELBO (shaded grey area). These solutions fall on the line of lowest limiting values $D = H(\pi) - R$, with $H(\pi)$ the stimulus entropy, such that both inequalities in Eq (24) are achieved; as a result, the mutual information is equal to $\bar{R}$ (Fig 4A, inset). Deviations from this line appear for extreme values of the target rate. On the one hand, as the stimulus and the generative distributions do not belong to the same parametric family, it is not possible to achieve the limiting value of the distortion with $\bar{R} = 0$ (this can be achieve by using a log-normal decoder, S2 Fig). On the other hand, for sufficiently large $\bar{R}$, the distortion stops decreasing and saturates; this occurs when the tuning curves are as narrow as possible while still tiling the stimulus space (Fig 4B, bottom). The distortion can be further decreased by increasing the number of available activity patterns, which depends on the population size (Fig 5A). By increasing the number of neurons, we also increase the number of tuning-curve arrangements which correspond to an optimal model, i.e., the number of degenerate solutions. This degeneracy is stronger at lower values of the rate, implying that the lowest limiting value of distortion can be achieved even with small population sizes. This is because, once the critical size needed to achieve the lowest limiting value of the distortion is reached, adding neurons does not raise the informational content of the population activity. When the rate is large, instead, the lowest limiting value can be achieved only in large enough populations.

The quality of the generative model is quantified by the $D_{KL}$ divergence between the generative distribution, $p_\psi(x)$, and the stimulus distribution, $\pi(x)$; it is negligible for all values of $\bar{R}$ in the region of interest (Fig 4B). (We recall that the ELBO, up to a constant, is a lower bound to this quantity, and the gap is the $D_{KL}$ divergence between the true and the approximate posterior distribution over neural activity, Eqs (9) and (10).) The U-shape is due to the jaggedness of the generative model at high values of $\bar{R}$ (see next section), which is attenuated as the population sizes increases (Fig 5B).

Different values of $\bar{R}$ also result in different encoders, corresponding to different arrangements of the tuning curves (Fig 4C). For small values of $\bar{R}$, tuning curves are broad and the spacing between preferred positions is small, causing large overlaps: different stimuli are

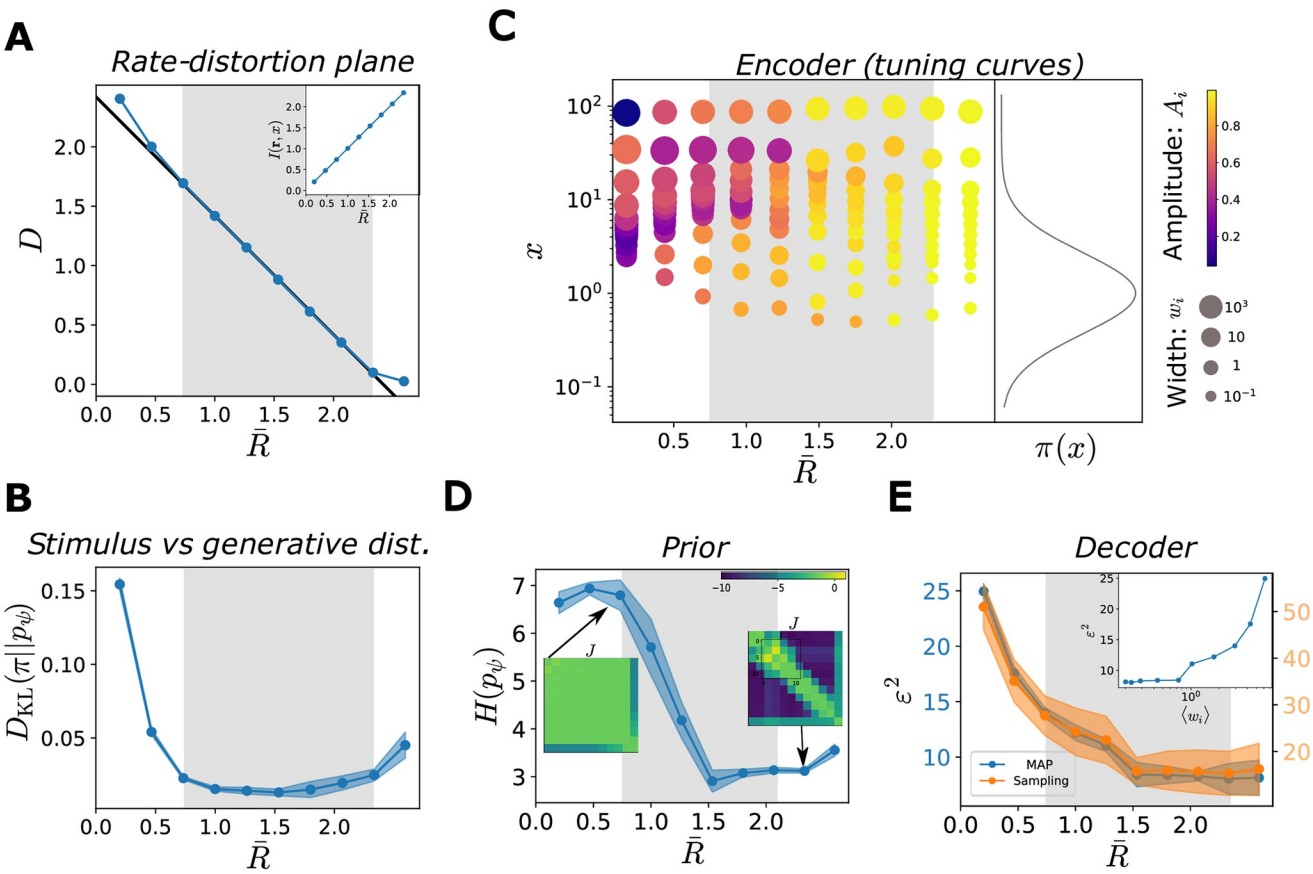

**Fig 4. Characterization of the optimal solutions as functions of the target rate.** In all simulations, $N = 12$, $\pi(x) = \mathcal{LN}(1,1)$. Solid curves illustrate the mean across different initializations and shaded regions correspond to one standard deviation. (**A**) Solutions of the ELBO optimization problem as a function of target rate, $D(\bar{R})$ (blue curve), and theoretical optimum, $D = H(\pi) - \bar{R}$ (black curve), in the rate-distortion plane. Values of $\bar{R}$ where the solutions coincide with the theoretical optimum (grey region). Solutions depart from the optimal line when the rate is very low (poor generative model) or very high (saturated distortion). Inset: mutual information between stimuli and neural responses as a function of $\bar{R}$. (**B**) Kullback-Leibler divergence between the stimulus and the generative distributions, as a function of $\bar{R}$. (**C**) Optimal tuning curves for different values of $\bar{R}$. Each dot represents a neuron: the position on the $y$-axis corresponds to its preferred stimulus, the size of the dot is proportional to the tuning width, and the color refers to the amplitude (see legend). Tuning curve parameters are averaged across 16 initializations, ordering the neurons as a function of their preferred position. The curve on the right illustrates the data distribution, $\pi(x)$. (**D**) Entropy of the prior distribution over neural activity, $p_\psi(\mathbf{r})$, as a function of $\bar{R}$. Insets show two configurations of the coupling matrices, with rows ordered according to the neurons' preferred stimuli, and coupling strengths colored according to the legend. (**E**) MSE in the stimulus estimate, obtained as the MAP (blue curve, scale on the left $y$-axis), or from samples (orange curve, scale on the right $y$-axis), as a function of $\bar{R}$. Inset: MSE (MAP) as a function of the average tuning width.

mapped to similar distributions over neural activity patterns. Moreover, they are characterized by low amplitudes and, thus, higher stochasticity; indeed, stochastic neurons yield compressed representations [50]. Increasing $\bar{R}$ causes noise to be suppressed through an increase in the amplitude, and narrower and more distributed tuning curves.

The solutions also differ in the structure of the prior over neural activity, $p_\psi(\mathbf{r})$ (Fig 4D, insets). In the regime in which the decoder ignores the latent variables, i.e., $p_\psi(x|\mathbf{r}) \approx p_\psi(x)$, the prior, $p_\psi(\mathbf{r})$, is unstructured, and the couplings, $J$, are weak. By contrast, when $\bar{R}$ is large, the structure of the stimulus distribution affects the coupling matrix in the prior, inducing coupling strengths that depend on the distances between the neurons' preferred positions. (The capacity of the model to reach the lowest limiting value of the distortion for a large range of target rates is precisely due to the prior being very flexible and able to capture correlations

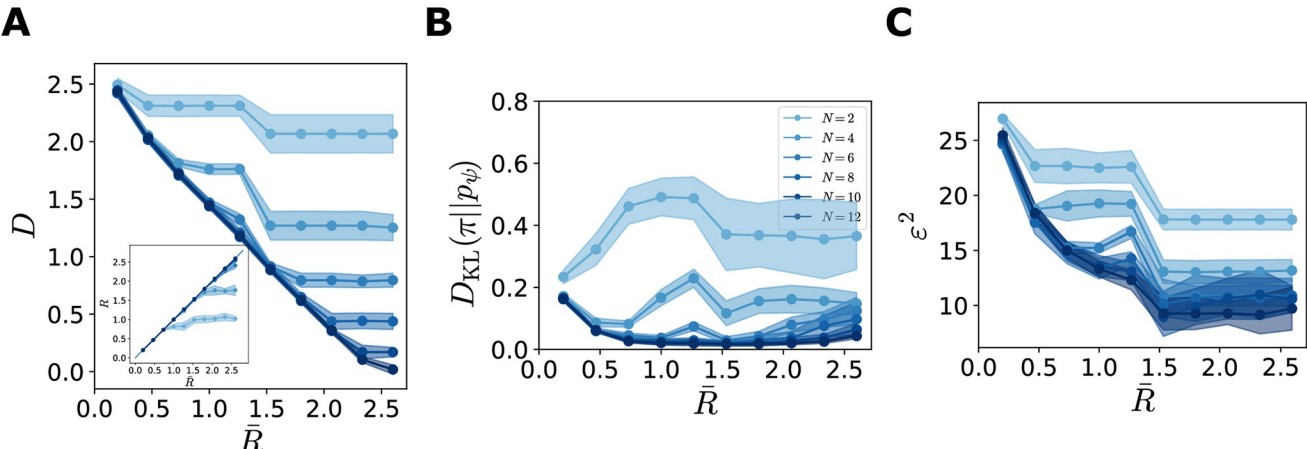

**Fig 5. Dependence of the results on the population size.** Solid curves illustrate the mean across different initializations and shaded regions correspond to one standard deviation. Same simulations as in Fig 4, with different values of the population size. (**A**) Optimal solutions (blue curves), $D(\bar{R})$, for different population sizes, $N$, (legend in panel B) and theoretical bound (black curve), $D = H(\pi) - \bar{R}$, in the rate-distortion plane. (**B**) Kullback-Leibler divergence between the stimulus and the generative distributions, as a function of $\bar{R}$. (**C**) MSE in the stimulus estimate, obtained as the MAP, as a function of $\bar{R}$.

between neural activity. The flexibility of the model is reduced when the couplings in the prior are set to 0 (S3 Fig).) As the coupling strengths increase, the entropy of the prior distribution decreases (Fig 4D).

Finally, we characterize the decoding properties in terms of a quantity commonly used in perceptual experiments and theoretical analyses: the mean-squared error (MSE) in the stimulus estimate. We approximate the maximum a posteriori (MAP) estimate as the mode of the decoding distribution (see Materials and methods). In the absence of an auxiliary, ideal estimator, the brain can produce estimate by sampling from the decoder; we thus also consider the MSE in the limiting case in which the stimulus estimate is evaluated as a single sample from the posterior distribution, $\hat{x}_{sampling} \sim p_\psi(x|\mathbf{r})$. In Materials and methods, we compute the two corresponding functional forms, (see Eqs (16)–(18)), which differ by a term equal to the posterior variance in the case of Gaussian decoder. We note that our decoder is not ideal, especially at low rates, and might be biased; it does not saturate the Cramer-Rao bound. As a consequence, as in the cases of extremely noisy neurons and complex encoding schemes [51–53], the Fisher information of the encoding distribution yields a poor estimate of the decoding accuracy (see Materials and methods, S6 Fig).

As expected from the behavior of the mutual information between stimuli and neural responses, the decoding performance of the system increases as a function of $\bar{R}$, with a similar qualitative behavior of the error in the two estimation schemes (Fig 4E). But it is worth examining the behavior quantitatively. In both schemes, the MSE does not decrease linearly with $\bar{R}$, but rather exhibits a rapid decrease followed by a slower one; the quantitative value at high rates depends on the population size (Fig 5C). In particular, the system achieves comparable decoding performances for a broad range of values of the tuning width (Fig 4E, inset).

## Generalization and the role of the rate as regularizer

The result illustrated in Fig 4B implies that intermediate representations are preferred to representations with extremely narrow tuning curves, in that they yield a smoother approximation of the stimulus distribution. This observation suggests that, in the case of a limited volume of data, the rate might serve as regularizer which prevents the model from overfitting the data.

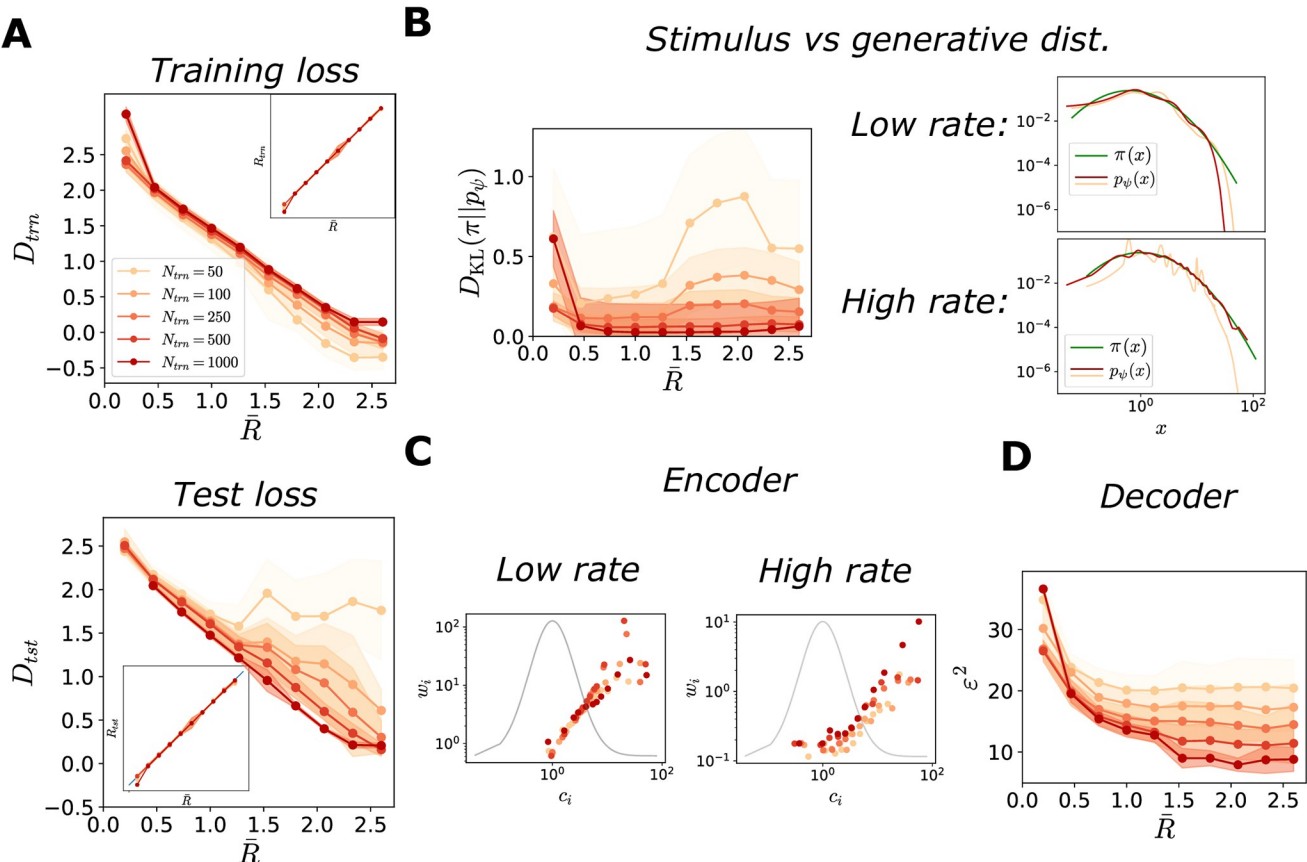

**Fig 6. Characterization of optimal solutions as functions of training set size.** In all simulations, $N = 12$, and $\pi(x) = \mathcal{LN}(1,1)$. Solid curves represent the mean across different initializations, and shaded regions correspond to one standard deviation. The legend in panel A serves as a legend for all panels. (**A**) Solutions of the ELBO optimization problem as functions of the target rate, for the training set (top) and for the test set (bottom). Top: distortion, $D_{trn}(\bar{R})$, and rate, $R_{trn}(\bar{R})$ (inset), for the training set as a function of the target rate, for different sizes of the training set, colored according to the legend. For smaller training sets, at higher rates the model tends to overfit the data, resulting in a lower training distortion than optimal (red line, large training set, same data as in Fig 4). Bottom: distortion, $D_{tst}(\bar{R})$, and rate, $R_{tst}(\bar{R})$ (inset), for the test set as functions of the target rate, for different sizes of the training set. For smaller training sets, at higher rates the model does not generalize to unseen samples, resulting in a large distortion. (**B**) Left: Kullback-Leibler divergence between the stimulus and the generative distributions, as a function of $\bar{R}$, for different sizes of the training set. At higher rates, the generative model fits poorly the stimulus distribution. Right: examples of comparisons between stimulus (green line) and generative distribution (red and orange line) at low (top) and high (bottom) rates, for different sizes of the training set, $N_{trn} = 100$ and $N_{trn} = 2000$, colored according to the legend as in panel A. (**C**) Tuning width, $w_i$, as a function of the location of a preferred stimulus, $c_i$ (dots), at low (left) and high (right) rates, for different sizes of the training set, $N_{trn} = 100$ and $N_{trn} = 1000$. The grey curve represents the stimulus distribution, $\pi(x)$. (**D**) MSE in the stimulus estimate, obtained as the MAP, as a function of $\bar{R}$, for different sizes of the training set.

We thus expect that limiting the rate benefits generalization properties, i.e., suppresses errors due to the stochasticity of the training dataset.

We test this hypothesis by training the model on a limited volume of data, and comparing the resulting performance to the asymptotic limit of a large training set explored in the previous section (Fig 6). We find that, at intermediate values of target rate (grey region in Fig 4), the distortion in the training set is always smaller than the distortion in the asymptotic limit, and the more so the smaller the training set, suggesting that the model overfits (Fig 6A, top). Moreoever, this trend is enhanced for larger values of the rate. This is further confirmed by an analysis of the distortion in a test set (a large number of stimulus samples), which is larger than in the asymptotic limit, yielding a 'generalization gap' that depends on the value of the target rate

(Fig 6A, bottom). The rate, instead, exhibits no difference between the training and the test set (Fig 6A, insets).

The generalization gap results in a poorer fit of the stimulus distribution by the generative model (Fig 6B). When the number of training samples is small, at high rates the location of the peaks in the generative distribution (see also Fig 3B, bottom) strongly depend on the stimuli present in the training set; this leads to a poor approximation of the stimulus distribution (Fig 6B, bottom). Low rates, instead, induce a smoother generative distribution. They also reduce the dependence of the quality of fit on the training set size, and enhance generalization (Fig 6B, left). Thus, the target rate effectively acts as regularizer, that prevents overfitting and favors generalization. A limitation in the volume of training data also affects the encoder and decoder properties: tuning curves become narrower at higher rates (Fig 6C), and the MSE in the stimulus estimate is much more sensitive to the volume of the training set at high rates (Fig 6D). We verify the consistency of these observations across other stimulus distributions by examining the case of a multimodal distribution (S5 Fig). Jointly, the results illustrated in Figs 4 and 6 point to the benefit of intermediate values of the rate that regularize the model. In this regime, the encoder is characterized by broad tuning curves, and the decoder achieves low coding error and generalizes to unseen data (Figs 4E and 6D).

## Optimal allocation of neural resources and coding performance

The classical efficient coding hypothesis prescribes an allocation of neural resources as a function of the stimulus distribution: more frequent stimuli are represented with higher precision. This has been proposed as an explanation of a number of measurements of perceptual accuracy and behavioral bias [10, 54, 55]. We investigate, in our model, the relations between stimulus distribution, the use of neural resources (tuning curves), and the coding performance, and how each these vary with $\bar{R}$. We emphasize that the functional form of the stimulus distribution affects these relations, through its interplay with the functional form not only of the encoder (as in the classical efficient coding framework), but also of the generative distribution. In order to make statements about the typical behavior of the system, we average our results over different random initializations of the parameters; single solutions might deviate from the average behavior due to the small number of neurons and the high dimensionality of the parameter space. We illustrate results for the non-trivial case of a log-normal distribution of stimuli and a Gaussian decoder; in S7 and S8 Figs, we report results obtained when the stimulus and decoding distributions belong to the same parametric family (Gaussian and log-normal, respectively). Our conclusions can be compared with results from previous studies. In particular, we invoke the analytical results derived in Ref. [10] for a similar population coding model; in Sec. S3 Appendix, we provide an alternative derivation of these results and we comment on the main differences with our model. Here, we note that our results are obtained by considering a regime of strong compression of the information (small population sizes), while previous studies focused on the asymptotic regime with $N \to \infty$.

As illustrated in Fig 4C, the target rate affects the neural density, i.e., the number of neurons with preferred stimuli within a given stimulus window. In previous work, maximizing the mutual information required that the neural density be proportional to the stimulus density, $d(x) \propto \pi(x)$ [10, 31, 56]. In our case, the range of possible behaviors is richer, especially when the stimulus distribution is non-trivial (i.e., it does not have the same functional form as that of the generative distribution). At low rates, the location of maximum density might be different from the mode of the stimulus distribution, depending on the interplay between the generative and the stimulus distributions (Fig 7A, S7(A) and S8(A) Figs). The neural density becomes more sensitive to the stimulus distribution for large values of $\bar{R}$: a power law

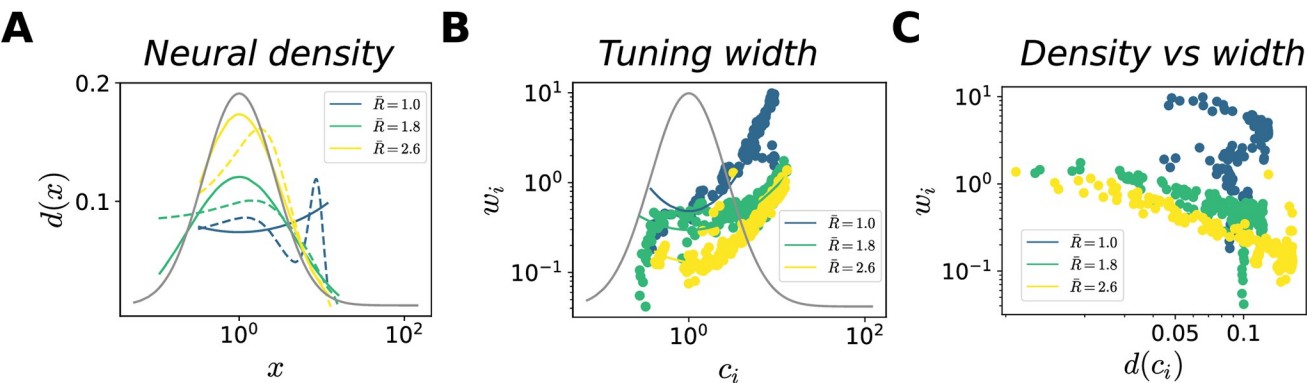

**Fig 7. Optimal allocation of neural resources.** In all simulations, $N = 12$ and $\pi(x) = \mathcal{LN}(1, 1)$. Results are illustrated for regions of the stimulus space where the coding performance is sufficiently high, defined as the region where the MSE is lower than the variance of the stimulus distribution. Below, we mention exponents of the power law fit when the variance explained is larger then a threshold, $R^2 \geq 0.7$. (**A**) Neural density as a function $x$ (dashed curves) and power-law fits (solid curves, $R^2 = (0.21, 0.83, 0.95)$, $\gamma_d = (-, 0.43, 0.62)$), for three values of $\bar{R}$ (low, intermediate, and high); the grey curve illustrates the stimulus distribution. The density is computed by applying kernel density estimation to the set of the preferred positions of the neurons. (**B**) Tuning width, $w_i$, as a function of preferred stimuli, $c_i$ (dots), and power-law fits (solid curves, $R^2 = (0.78, 0.42, 0.82)$, $\gamma_w = (1.15, -, 0.7)$) for three values of $\bar{R}$; the grey curve illustrates the stimulus distribution. (**C**) Tuning width, $w_i$, as a function of the neural density, $d(c_i)$, for three values of $\bar{R}$; Pearson correlation coefficient $\rho = (-0.74, -0.66, -0.79)$.

functional form, $d(x) = A_d \pi(x)^{\gamma_d}$, yields a good agreement with our numerical results, with an exponent, $\gamma_d$, close to 1/2 (Fig 7A).

In Ref. [10, 57], analytical results were obtained by constraining the neural density and the tuning width relative to each other. This is equivalent to fixing the overlap between tuning curves, by imposing $w(x) \propto d^{-1}(x) \propto \pi(x)^{-1}$ (see Sec. S3 Appendix). In our case, the tuning width and neural density vary independently of each other, and the distribution of widths exhibits an intricate behavior at small values of $\bar{R}$ (Fig 7B, S7(B) and S8(B) Figs). At large values of $\bar{R}$, the tuning width decreases for large values of the stimulus distribution, and its behavior is well described by a power law, $w_i = A_w / p(c_i)^{\gamma_w}$. As a result, as $\bar{R}$ increases, the inverse correlation between the neural density and the tuning width becomes sharper (Fig 7C and S7(C) Fig).

A consequence of the heterogeneous allocation of neural resources is a non-uniform coding performance across stimuli. Fig 8A shows that the MSE exhibits an inverse relation as a function of the stimulus distribution, with more frequent stimuli encoded more precisely. This is broadly consistent with previous studies [10, 58], which maximized the mutual information to obtain the expression

$$\varepsilon^2(x) \propto \frac{1}{p^2(x)}. \tag{27}$$

(Similar power-law behaviors, with different exponents, arise from different loss functions [10, 12, 59, 60].) Concretely, these power-law expressions were derived using the Fisher information, whose inverse is a lower bound to the variance of an unbiased estimator, and which can be related to the mutual information in some limits. Here, for all values of $\bar{R}$, the error is well described by a power law, $\varepsilon^2(x) = A_\varepsilon / \pi(x)^{\gamma_\varepsilon}$, where the exponent changes as a function of $\bar{R}$, and depends on the choice of the decoder (Fig 8A, S7(D) and S8(D) Figs). In particular, we find that when the decoder and the generative distributions belong to the same parametric family, the dependence of the error on the stimulus distribution is stronger, characterized by larger values of $\gamma_\varepsilon$. Finally, we illustrate how the configuration of the tuning curves affects the

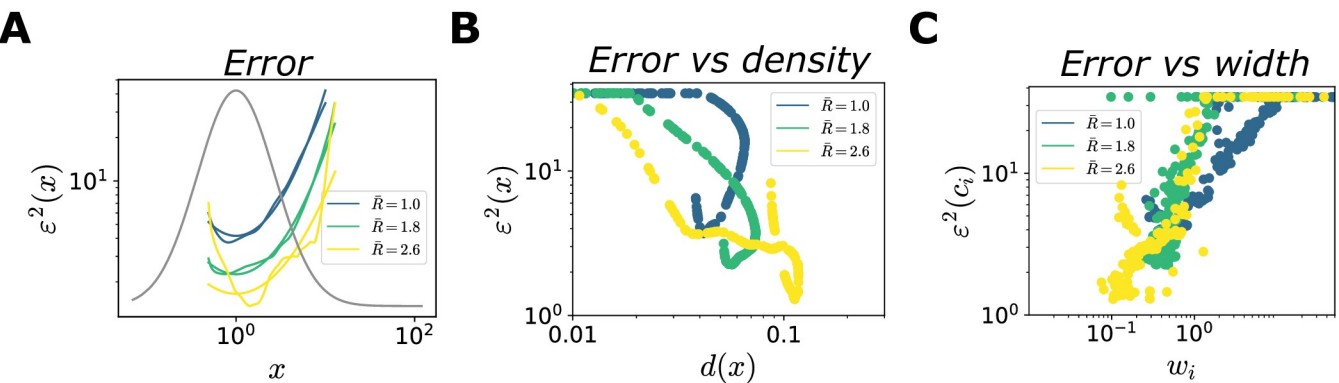

**Fig 8. Optimal allocation of coding performance.** Same numerical simulations as in Fig 4. (**A**) MSE (MAP estimate) as a function of $x$ (dashed curves), and power-law fits (solid curves, $R^2 = (0.98, 0.98, 0.76)$, $\gamma_e = (0.87, 0.74, 0.59)$), for three values of $\bar{R}$. (**B**),(**C**) MSE as a function of the neural density (B) and tuning width (C), for three values of $\bar{R}$; Pearson correlation coefficient $\rho_{density} = (-0.66, -0.96, -0.81)$, $\rho_{width} = (0.36, 0.59, 0.70)$.

coding performance, by plotting the MSE as a function of the neural density and tuning width. We observe a correlation between high coding performance and regions of high neural density and narrow tuning widths (Fig 8B and 8C, S7(E), S7(F), S8(E) and S8(F) Figs).

To summarize, given our choice of the loss function, which constrains the encoding stage as a function of the decoding stage, we obtain a family of optimal neural representations. In weakly constrained systems (large values of $\bar{R}$), we qualitatively recover previously derived relationships between tuning curves, stimulus distribution, and coding performance. (The difference in the numerical values of the exponents in the power laws can be explained by the differences between the models (see Sec. S3 Appendix), and depends on the similarity between the functional forms of the generative and the stimulus distributions. We note that, in previous work [10, 12, 59] the numerical value of the exponents also change as a function of the form of the loss function.) In systems with stringent information compression (small values of $\bar{R}$), the optimal resource allocation exhibits a more intricate behavior that depends on the functional form of the stimulus distribution and the properties of the generative model.

## Case study: Neural encoding of acoustic frequencies

We validate our theory on existing data by considering the empirical distribution of acoustic frequencies in the environment and relating it to behavioral performance. This distribution was obtained in Ref. [57] by fitting the power spectrum of recordings data, $S(f)$, with a power-law,

$$S(f) = \frac{A}{f_0^p + f^p}, \tag{28}$$

with $A = 2.4 \times 10^6$, $f_0 = 1.52 \times 10^3$ and $p = 2.61$ (Fig 9A, inset). Since the stimulus distribution exhibits a heavy-tail, we test our model with both a Gaussian and a log-normal decoder. Despite exhibiting similar degeneracy in the space of solutions (Fig 4 and S2 Fig), the two decoders yield different quantitative predictions on the optimal allocation of neural resources (Figs 7 and 8, and S8 Fig).

We observe that a broad range of values of $\bar{R}$ results in comparable values of the ELBO and comparable generative model performances (Fig 9A and 9D). A log-normal decoder yields a better fit of the stimulus distribution as it is better suited than a Gaussian to capture the heavy

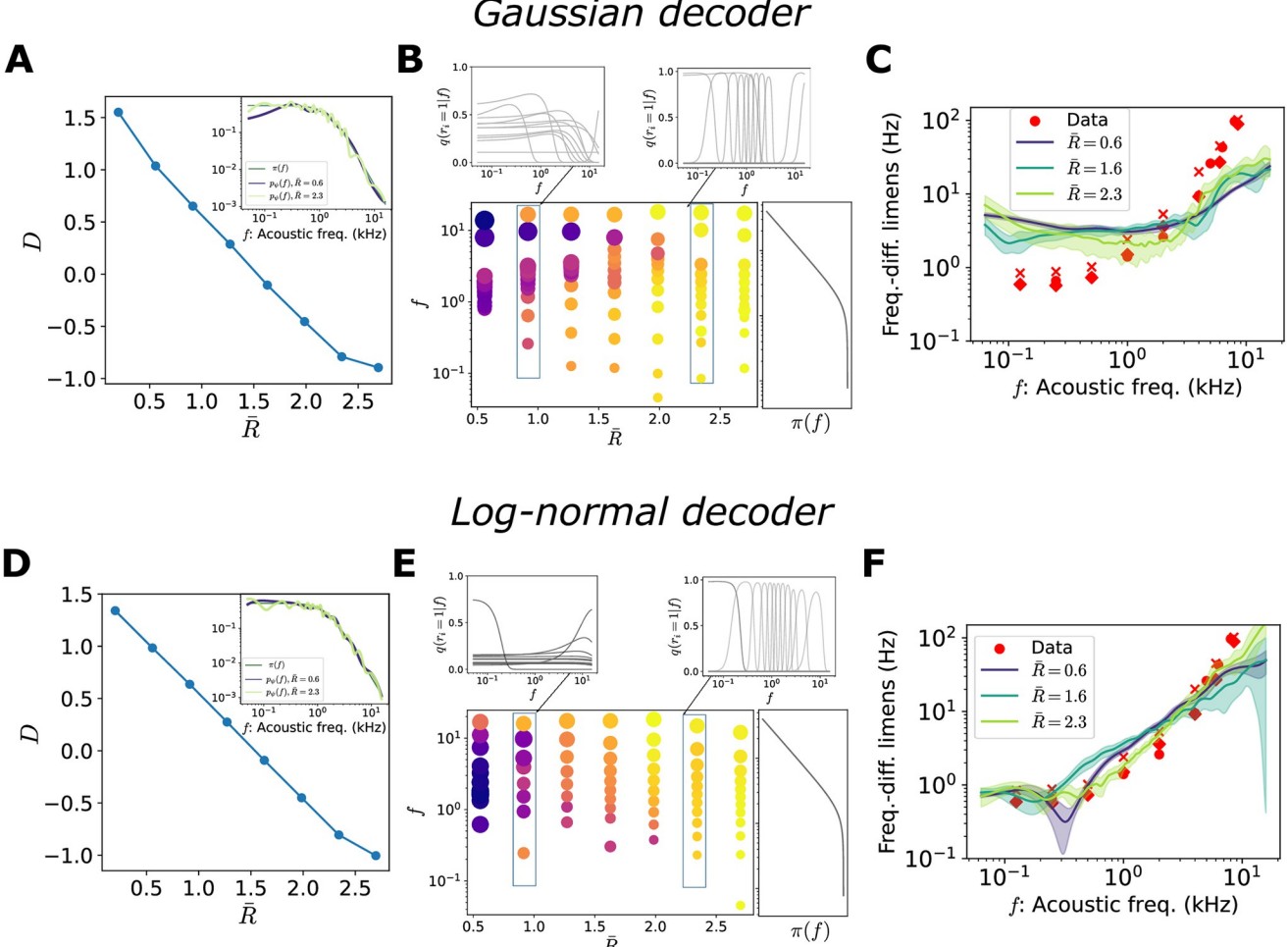

**Fig 9. Generative models for the distribution of acoustic frequencies.** In all simulations, $N = 12$. The decoder is either Gaussian (top row) or log-normal (bottom row). (**A**) Solutions of the optimization problem as a function of the target rate, $D(\bar{R})$ (blue curve), in the rate-distortion plane. Inset: environmental distribution of acoustic frequencies, $\pi(f)$, and generative model fit for two different values of the target rate, colored according to the legend. (**B**) Optimal tuning curves for different values of $\bar{R}$. Each dot represents a neuron: the position on the $y$-axis corresponds to its preferred stimulus, the size of the dot is proportional to the tuning width, and the color refers to the amplitude (see legend in Fig 4). The curve on the right illustrates the stimulus distribution, $p(f)$. Insets show two examples. (**C**) Frequency discrimination as a function of acoustic frequency. Red markers are data points from three different subjects, data from Ref. [61]. Solid curves are the RMSE for three values of $\bar{R}$, scaled by a factor of $\hat{a} = (281, 142, 107)$, with variance explained $R^2 = (0.42, 0.41, 0.66)$. (**D**)-(**F**) Same as panels (A)-(C) in the case of a log-normal decoder. In panel F, $\hat{a} = (187, 104, 72)$, with variance explained $R^2 = (0.92, 0.81, 0.96)$. Solid curves illustrate the mean across different initializations and shaded regions correspond to one standard deviation.

tail in the stimulus distribution (Gaussian decoder $D_{KL} \approx 0.1$, log-normal decoder, $D_{KL} \approx 0.04$). As before, for increasing $\bar{R}$, solutions are characterized by an encoder with increasingly narrow tuning curves and the locations of preferred stimuli sensitive to stimulus probability (Fig 9B and 9E).

Finally, we test the prediction of our model regarding the dependence of the error on the stimulus value by comparing it to experimental data. We borrow experimental measurements of the so-called frequency-difference limens, the minimum detectable changes in the frequency of a sinusoidal sound wave, from Ref. [61]. We employ the square root of the MSE (RMSE) of the stimulus estimate as a proxy for perceptual resolution. Since the small number of neurons imposes a fundamental bound to the coding performance, we scale the RMSE by a

constant factor, $\hat{a}$, which can be thought of as a population size gain, to allow for a comparison (see Materials and methods).

In the case of a Gaussian decoder, the functional form of the RMSE does not capture the behavior of the frequency-difference limens for any value of the target rate (Fig 9C). By using a log-normal decoder, instead, we obtain a faithful description of the frequency-difference limens for a broad range of values of $\bar{R}$ (Fig 9F). These results show that, despite a large variability in the parameters of the encoder, as is commonly observed in biological systems, robust predictions in the perceptual domain can be obtained and are consistent with experimental data. In future work, it would be interesting to conduct a more systematic study on the aspects of the generative model, e.g., properties of the functional form of the decoder, that can be extracted from data.

## Discussion

We studied neural representations that emerge in a framework in which populations of neurons encode information about a continuous stimulus with simple tuning curves, but with the additional assumption that the task of the decoder is to maintain a generative model of the stimulus distribution. The consequence of the specific task imposed on the *decoder* is that the *encoder* is set so as to maximize a bound to the mutual information between stimulus and neural activity, as postulated by the efficient coding hypothesis, subject to a constraint on the relative entropy between evoked and prior distributions over neural activity.

Under this constraint, different optimal solutions are obtained, corresponding to equally accurate generative models but (qualitatively) different neural representations of the stimulus (Fig 3). These representations differ in the degree of compression of information in the neural responses, reflected in encoding (neural) properties (Figs 4 and 7), in the generative model prior over neural activity (Fig 4D), in the generalization properties of the model (Fig 6), and in the coding performance (Figs 4 and 8). For intermediate degrees of compression, the optimal model is characterized by broad tuning curves and exhibits low coding error and an accurate generative model with robust generalization properties. We emphasize that optimal solutions are learned from a set of stimulus samples, and do not require full knowledge of the stimulus prior as opposed to classic efficient coding models.

### VAE, efficient coding, and learning from examples

The VAE can be viewed as a generalized efficient coding framework. The encoder is jointly optimized with the decoder to maximize a variational approximation of the mutual information under a constraint on neural resources. However, unlike classical efficient coding frameworks in which the functional forms of the encoding accuracy and the resource constraint can be chosen with a certain degree of arbitrariness [10–12, 30, 58, 59], here they are constrained by a unique objective function that links them together. Thus encoding and decoding are captured in a unified approach [62]. In previous models, the form of the predictions, such as power laws governing the dependence of the coding performance on the stimulus probability density depended on the choice of the two terms in the loss function. Different combinations of alternative choices of these two terms can yield the same prediction [12, 59]. Here, instead, flexibility is achieved through the choice of the functional form of the generative model and the bound on the resource constraint. Furthermore, our model exhibits broad tuning curves for intermediate values of the target rate without further constraints (see section "Optimal tuning width" and S3 Appendix).

Another difference with the classic efficient coding framework is that, here, the decoder does not make use of the knowledge of the distribution of stimuli to perform: no 'Bayesian

inversion' is needed. The benefit of an approximate Bayesian inversion (through a variational approximation of the 'inverted encoder'), is that solutions can be learned on the basis of data samples [26, 27, 63]. This is an advantage of the VAE framework, as in typical natural situations the stimulus distribution is accessible only through observations.

## VAEs in neuroscience: Related studies

VAEs are among the state of the art approaches to unsupervised learning, and in recent years they have been applied in different contexts in neuroscience to model neural responses. Several studies have considered neuroscience-inspired VAEs, in which the generative model was based on a decomposition of natural images into sparse combinations of linear features [64]. It was then paired with a powerful encoder, which models the sensory encoding process, and specific assumptions on the prior distribution of the latent variables, to obtain representations similar to the ones observed in the early visual pathway (in V1 and V2) [18, 26, 65]. In these models, the simplicity of the generative distribution prevented posterior collapse. We note that, in our case, we reverse this approach, by assuming a specific a simple and biologically motivated form of the encoder (a set of tuning curves), while we allow for a flexible decoder.

In the context of higher visual areas instead, more complex generative models were needed to capture neural representations [66]; to overcome the issue of posterior collapse, the authors used a loss function akin to the one in Eq (22), but the value of $\beta$ was chosen by hand. In doing so, they obtained an empirical advantage in the semantic interpretability of the latent variables, at the cost of abandoning the requirement that the loss function be a bound to the log-likelihood. This, so-called, $\beta$-VAE approach was also employed in Ref. [67] to study optimal tuning curves in a population coding model of spiking neurons similar to ours. In this study, however, the population as a whole was constrained to emit one spike only, limiting the number of available activity patterns to $N$ (the number of neurons). Moreover, the encoder and the decoder are not optimized independently; this choice prevented the emergence of multiple alternative neural representations in the $\beta = 1$ case. By varying $\beta$, the authors obtained neural representations which differed in the shape of the optimal tuning curves, but, since for $\beta \neq 1$ the ELBO was not optimized, the resulting generative model was not accurate.

## What causes the degeneracy of the optimal solutions?

Degeneracy in the space of solutions results from the flexibility of the generative model. In our case, the generative model is a mixture of distributions: we restrict our choice to two examples of parametric families, Gaussian and log-normal. Despite their simplicity, the generative model is equipped with high approximation capabilities. Indeed, the marginal distribution, $p_\psi(x) = \sum_{\mathbf{r}} p_\psi(x|\mathbf{r}) p_\psi(\mathbf{r})$, is a mixture of distributions, and, in the case of Gaussian decoder, is a universal approximator of densities (i.e., a well-chosen Gaussian mixture can be used to approximate any smooth density function [40, 41]).

With a population of $N$ binary neurons the generative model consists, in principle, of a mixture of $2^N$ different distributions. The association between each activity pattern and the mean and variance of mixture components is determined by the decoder, while the prior on the neural activity constrains how many distributions are effectively employed, as it governs the mixing factors. At low rates, the prior distribution is not constrained by the encoder and can exhibit high entropy. At low distortion, instead, latent variables are mapped to distinct output distributions. The prior distribution over latent variables is constrained to capture the correlations in neural activity imposed by narrow tuning curves; its entropy is thus suppressed, resulting in a limited number of non-vanishing mixing factors (and the performance of the model degrades if the prior is not flexible enough to capture these correlations, S3 Fig). Intermediate

rates balance these two limits, yielding a generative model with a sufficient number of mixing factors and distinct output distributions, which reproduces the stimulus distribution and generalizes to unseen data [68–70].

Here, we focused on relatively simple, one-dimensional stimulus distributions. As the statistics of many natural features are dominated by low-frequency components (e.g., spatial frequencies in natural images), and if powerful decoders are to represent deep brain areas [18, 66], we expect degeneracy in the space of solutions even in the case of multi-dimensional stimuli. In future work, it would be interesting to pair powerful decoders with biologically inspired high-dimensional encoders, e.g., multidimensional tuning curves, and to characterize the degeneracy of the solutions. In the case of more complex stimuli, the rate will represent a more stringent condition in controlling the generalization properties of the model [69, 70].

## Internal models and perception as inference

Our choice on the form of the decoder stems from the assumption that organisms interact with their environment with the use of internal models. These allow them to perform inference and make predictions. But what form do internal models take and what is their neural substrate? In previous studies [17, 21, 22, 71], internal models were defined by conditioning the probability of a stimulus, $x$, on the realization of a latent variable, $z$, through their joint distribution, $p(z, x) = p(x|z)p(z)$. The latent variables were chosen so as to allow for a semantic interpretation, such as the presence or absence of a given image feature (e.g., Gabor patches or texture features [18]). Sensory areas were then assumed to compute a posterior distribution over the latent variable, $q(z|x)$, and the neural activity was invoked as a way to represent this posterior distribution, either approximately from samples [17, 22, 72], or as parameters of a parametric distribution [19, 20].

We chose to define the generative model directly as a joint distribution of two random variables, $p(\mathbf{r}, x)$; $\mathbf{r}$ is the neural activity, while $x$ is defined on the space of stimuli. Although we preserve the mathematical structure of generative models proposed in neuroscience (e.g., the Helmoltz machine and its more recents extensions [14, 19]), our interpretation is different. The neural activity itself plays the role of a latent representation of the stimulus, but it is not set, a priori, to some interpretable feature, such as the presence or the intensity of a Gabor filter in models involving natural images (as in Refs. [17, 22]). In order to constrain sensory areas, we assume the generative model to be implemented in downstream areas and we model its output with a flexible function, a neural network, which outputs a point estimate and an uncertainty about the value of the stimulus [39, 73]. This output corresponds to a perceptual representation of the stimulus in the brain, and can be related to behavioral measurements (as in Fig 9C).

Mathematically, the encoding distribution, $q_\theta(\mathbf{r}|x)$, is obtained as a variational approximation of the posterior distribution of the generative model, $p_\psi(\mathbf{r}|x)$, as in previous work. This distribution, however, is defined on the space of neural activity patterns, and not on a set of abstract features. This choice has the drawback of the absence of a simple semantic interpretation of the latent features, but presents the advantage of a natural connection with an encoder based on properties of a neural system, e.g., a set of tuning curves and a model of neural noise. In the case of flexible generative models, different statistics of the latent variables turn out to be optimal. In this sense, the choice of the encoder, as well as the prior of the generative model, is useful to impose a structure on the characteristics of the neural representations.

## Optimal tuning width

Our choice of encoding model allows us to compare our results with those of earlier studies that considered the optimal arrangement of neurons with bell-shaped tuning curves in the

presence of non-uniform stimulus distributions [10, 31]. While for higher values of the target rate we recover the previously derived allocation of neural resources as a function of the stimulus distribution, the behavior for lower values of the target rate is more intricate, and depends on the specifics of the stimulus distribution. Thus, in our case, the constraint on neural resources has a stronger impact on their optimal allocation than, for example, in Ref. [10], where the bound on the mean activity in the population acts merely as a scaling factor, and the behavior of the tuning curves is more constrained. In particular, in Ref. [10] the tuning width was fixed a priori to be inversely proportional to the neural density, to enforce a fixed amount of overlap between tuning curves: it was not optimized. This choice was made to avoid a common issue in this type of calculations: in the case of a one-dimensional stimulus and in the asymptotic limit of infinitely many neurons, the maximization of the mutual information yields the pathological solution of infinitely narrow tuning curves [74, 75]. Metabolic constraints on the neural activity do not solve the issue, as narrow tuning curves can exhibit a moderate activity (as long as their amplitude is not too large).

In our framework, instead, the optimal tuning width and the amount of overlap between tuning curves are both optimized and vary as a function of $\bar{R}$. Moreover, a regime with intermediate values of the constraint, in which tuning curves are broad, exhibits both a smooth generative model (low $D_{\mathrm{KL}}$ divergence) *and* a low MSE (Fig 4B and 4E). Broad tuning curves are beneficial to obtain smooth generative models, while still allowing high for coding performance.

## Interpretation of the resource constraint

The constraint in Eq (20) involves the divergence between the evoked neural activity and its marginal distribution according to the generative model. This formulation is different from usual metabolic constraints which are designed to account for the energetic cost of neural activity [8]. In general, the cost should increase with the magnitude of the perturbation induced by a stimulus. Our more abstract formulation of the cost satisfies this property.

In our case, the prior distribution is parametrized by the biases and couplings of an Ising model. As we have shown, there are multiple ways to achieve a statistically optimal internal model and to minimize the $D_{\mathrm{KL}}$ divergence between the two sides of Eq (26), which differ in the value of the rate. At low rates, Eq (26) is approximated by relying on the optimization of the encoder parameters which are set so as to make $q_\theta(\mathbf{r}|x)$ similar to the prior for all stimuli; this then results in an unstructured coupling matrix in the prior distribution (Fig 4E, top). Conversely, at high rates, the encoder has a well defined structure which achieves a low distortion, and Eq (26) is approximated by optimizing the parameters of the prior and embedding the structure of the average posterior distribution in the connectivity matrix (Fig 4E, bottom). The value of the target rate can therefore be thought of as a cost of imposing structure in prior (spontaneous activity), through circuit properties. Thus, our model suggest an alternative normative principle to govern neural couplings as compared to information maximization, as proposed in Ref. [37].

A more concrete biological interpretation of the rate can be made by referring to the results obtained in Ref. [71], in which the prior distribution over latent variables of an internal model is related to the spontaneous neural activity. The authors start from the observation that in a well-calibrated internal model the prior equals the mean posterior, Eq (26) [15]. By comparing the average evoked activity to the spontaneous activity according to the $D_{\mathrm{KL}}$ divergence, the authors show that the two quantities become closer during development, and that this phenomenology is specific to naturalistic stimuli. This finding is then proposed as evidence of an internal model in primary visual cortex optimized for natural images, acquired gradually

during development. In this picture, the prior distribution of the generative model is identified with the spontaneous neural activity; we note, however, that there is no a priori reason to expect this relationship.

We can also relate information-theoretic quantities to biophysical processes by invoking results from statistical physics. A recent study has shown that the magnitude of the response of a system to an external perturbation is bounded above by the Kullback-Leibler divergence between the probability distributions describing the perturbed and unperturbed system [76]. Thus, if we view the marginal distribution as describing the unperturbed state of the neural population, the rate term provides an upper bound on the magnitude of the response, i.e., the change in firing rate, of the neural population upon a stimulus presentation. In turn, this quantity is proportional to the spiking metabolic cost. Ultimately, one would like to derive formulations of the abstract, information-theoretic costs that govern joint encoder-decoder models from known microscopic biophysical processes.

## Supporting information

**S1 Fig. Qualitatively different optimal configurations.** Same as Fig 3, in the case of lognormal distribution over stimuli, $\pi(x) = \mathcal{LN}(1, 1)$. Top row: high-distortion, low-rate solution. Bottom row: low-distortion, high-rate solution. (**A**) Bell-shaped tuning curves of the encoder (probability of neuron $i$ to emit a spike, as a function of $x$). (**B**) Comparison between the stimulus distribution, $\pi(x)$ (green curve), and the generative distribution, $p_\psi(x) = \sum_{\mathbf{r}} p_\psi(x|\mathbf{r})p_\psi(\mathbf{r})$ (purple curve). (**C**) Numerical values of the ELBO, and the distortion and rate terms.
(PDF)

**S2 Fig. Characterization of the optimal solutions as functions of the target rate in the case of same parametric family of stimulus and generative distribution.** Same as Fig 4, but with a log-normal decoder. (**A**) Solutions of the ELBO optimization problem as a function of target rate, $D(\bar{R})$ (blue curve), and theoretical optimum, $D = H(\pi) - R$ (black curve), in the rate-distortion plane. Values of $\bar{R}$ where the solutions coincide with the theoretical optimum (grey region). Since the decoder belong to the same parameteric family of the stimulus distribution, it is possible to achieve optimal distortion at very low rates. (**B**) $D_{\mathrm{KL}}$ divergence between the stimulus and the generative distributions, as a function of $\bar{R}$. (**C**) Optimal tuning curves for different values of $\bar{R}$. Each dot represents a neuron: the position on the $y$-axis corresponds to its preferred stimulus, the size of the dot is proportional to the tuning width, and the color refers to the amplitude (see legend). The curve on the right illustrates the data distribution, $\pi(x)$. (**D**) Entropy of the prior distribution over neural activity, $p_\psi(\mathbf{r})$, as a function of $\bar{R}$. Insets show two configurations of the coupling matrices, with rows ordered according to the neurons' preferred stimuli, and coupling strengths colored according to the legend. (**E**) MSE of the stimulus estimate, obtained as the MAP (blue curve, scale on the left $y$-axis), or from samples (orange curve, scale on the right $y$-axis), as a function of $\bar{R}$. Inset: MSE (MAP) as a function of the average tuning width.
(PDF)

**S3 Fig. Characterization of the optimal solutions as functions of the target rate in the case of less flexible prior on neural activity.** Same as Fig 4, but with $p_\psi(\mathbf{r})$ a product of independent Bernoulli distributions. The decoder is Gaussian. (**A**) Solutions of the ELBO optimization problem as a function of target rate, $D(\bar{R})$ (blue curve), and theoretical optimum, $D = H(\pi) - R$ (black curve), in the rate-distortion plane. Values of $\bar{R}$ where the solutions coincide with the theoretical optimum (grey region). Solutions always depart from the optimal line, especially at very high rate, due to the limited flexibility of the prior over neural activity. Inset: mutual

information between stimuli and neural responses as a function of $\bar{R}$. (**B**) $D_{\text{KL}}$ divergence between the stimulus and the generative distributions, as a function of $\bar{R}$. (**C**) Optimal tuning curves for different values of $\bar{R}$. Each dot represents a neuron: the position on the *y*-axis corresponds to its preferred stimulus, the size of the dot is proportional to the tuning width, and the color refers to the amplitude (see legend). The curve on the right illustrates the data distribution, $\pi(x)$. (**D**) Entropy of the prior distribution over neural activity, $p_\psi(\mathbf{r})$, as a function of $\bar{R}$. Insets show two configurations of the biases, *h*, as a function of the neuron preferred positions. (**E**) MSE of the stimulus estimate, obtained as the MAP (blue curve, scale on the left *y*-axis), or from samples (orange curve, scale on the right *y*-axis), as a function of $\bar{R}$. Inset: MSE (sampling) as a function of the average tuning width.
(PDF)

**S4 Fig. Characterization of the optimal solutions as functions of the target rate in the case of a multimodal distribution.** Same as Fig 4, but with $\pi(x)$ a multimodal distribution: a mixture of three Gaussians with means {−4, 0, 2}; variances {1, 0.5, 1}; and mixture coefficients {0.3, 0.2, 0.5}. The decoder is Gaussian. (**A**) Solutions of the ELBO optimization problem as a function of target rate, $D(\bar{R})$ (blue curve), and theoretical optimum, $D = H(\pi) − R$ (black curve), in the rate-distortion plane. Values of $\bar{R}$ where the solutions coincide with the theoretical optimum (grey region). Solutions depart from the optimal line when the rate is very low (poor generative model) or very high (saturated distortion). Inset: mutual information between stimuli and neural responses as a function of $\bar{R}$. (**B**) $D_{\text{KL}}$ divergence between the stimulus and the generative distributions, as a function of $\bar{R}$. Insets: two examples of comparison between stimulus (green curve) and generative distribution (purple curve). (**C**) Optimal tuning curves for different values of $\bar{R}$. Each dot represents a neuron: the position on the *y*-axis corresponds to its preferred stimulus, the size of the dot is proportional to the tuning width, and the color refers to the amplitude (see legend). The curve on the right illustrates the data distribution, $\pi(x)$. (**D**) Entropy of the prior distribution over neural activity, $p_\psi(\mathbf{r})$, as a function of $\bar{R}$. Insets show two configurations of the coupling matrices, with rows ordered according to the neurons' preferred stimuli, and coupling strengths colored according to the legend. (**E**) MSE of the stimulus estimate, obtained as the MAP (blue curve, scale on the left *y*-axis), or from samples (orange curve, scale on the right *y*-axis), as a function of $\bar{R}$. Inset: MSE (sampling) as a function of the average tuning width.
(PDF)

**S5 Fig. Characterization of optimal solutions as functions of training set size.** Same as Fig 6, but with $\pi(x)$ a multimodal distribution: a mixture of three Gaussians with means {−4, 0, 2}; variances {1, 0.5, 1}; and mixture coefficients {0.3, 0.2, 0.5}. The legend in panel A serves as a legend for all panels. (**A**) Solutions of the ELBO optimization problem as functions of the target rate, for the training set (top) and for the test set (bottom). Top: distortion, $D_{trn}(\bar{R})$, and rate, $R_{trn}(\bar{R})$ (inset), for the training set as a function of the target rate, for different sizes of the training set, colored according to the legend. For smaller training sets, at higher rates the model tends to overfit the data, resulting in a lower training distortion than optimal (red line, large training set, same data as in S4 Fig). Bottom: distortion, $D_{tst}(\bar{R})$, and rate, $R_{tst}(\bar{R})$ (inset), for the test set as functions of the target rate, for different sizes of the training set. For smaller training sets, at higher rates the model does not generalize to unseen samples, resulting in a large distortion. (**B**) Left: Kullback-Leibler divergence between the stimulus and the generative distributions, as a function of $\bar{R}$, for different sizes of the training set. At higher rates, the generative model fits poorly the stimulus distribution. Right: examples of comparisons between stimulus (green line) and generative distribution (red and orange line) at low (top) and high

(bottom) rates, for different sizes of the training set, $N_{trn}$ = 100 and $N_{trn}$ = 2000, colored according to the legend as in panel A. (**C**) Tuning width, $w_i$, as a function of the location of a preferred stimulus, $c_i$ (dots), at low (left) and high (right) rates, for different sizes of the training set, $N_{trn}$ = 100 and $N_{trn}$ = 2000. The grey curve represents the stimulus distribution, $\pi(x)$. (**D**) MSE in the stimulus estimate, obtained as the MAP, as a function of $\bar{R}$, for different sizes of the training set.
(PDF)

**S6 Fig. Comparison between Cramer-Rao bound and decoding error.** MSE (MAP estimate) (blue, green and yellow curves), and Cramer-Rao bound (inverse of the Fisher information, orange curves), as in Eq (19) as a function of $x$.
(PDF)

**S7 Fig. Optimal allocation of neural resources and coding performance.** Same as Figs 7 and 8, in the case of a Gaussian distribution $\pi(x) = \mathcal{N}(0, 5)$ (same of Fig 3) and Gaussian decoder. (**A**) Neural density as a function $x$ (dashed curves) and power-law fits (solid curves, $R^2$ = (0.96, 0.99, 0.99), $\gamma_d$ = (0.99, 0.71, 0.64)), for three values of $\bar{R}$ (low, intermediate, and high); the grey curve illustrates the stimulus distribution. The density is computed by applying kernel density estimation to the set of the preferred positions of the neurons. (**B**) Tuning width, $w_i$, as a function of preferred stimuli, $c_i$ (dots), and power-law fits (solid curves, $R^2$ = (0.09, 0.87, 0.92), $\gamma_w$ = (−, 0.77, 0.66)) for three values of $\bar{R}$; the grey curve illustrates the stimulus distribution. (**C**) Tuning width, $w_i$, as a function of the neural density, $d(c_i)$, for three values of $\bar{R}$; Pearson correlation coefficient $\rho$ = (0.30, −0.91, −0.97). (**D**) MSE (estimate obtained through sampling) as a function of $x$ (dashed curves), and power-law fits (solid curves, $R^2$ = (0.99, 0.98, 0.62), $\gamma_e$ = (1.37, 1.73, 1.86)), for three values of $\bar{R}$. (**E**),(**F**) MSE as a function of the neural density (E) and tuning width (F), for three values of $\bar{R}$; Pearson correlation coefficient $\rho_{density}$ = (−0.84, −0.90, −0.91), $\rho_{width}$ = (0.38, 0.79, 0.90).
(PDF)

**S8 Fig. Optimal allocation of neural resources and coding performance.** Same as Figs 7 and 8, in the case of log-normal decoder. (**A**) Neural density as a function $x$ (dashed curves) and power-law fits (solid curves, $R^2$ = (0.48, 0.94, 0.94), $\gamma_d$ = (−, 0.60, 0.63)), for three values of $\bar{R}$ (low, intermediate, and high); the grey curve illustrates the stimulus distribution. The density is computed by applying kernel density estimation to the set of the preferred positions of the neurons. (**B**) Tuning width, $w_i$, as a function of preferred stimuli, $c_i$ (dots), and power-law fits (solid curves, $R^2$ = (0.02, 0.90, 0.98), $\gamma_w$ = (−, 0.74, 0.74)) for three values of $\bar{R}$; the grey curve illustrates the stimulus distribution. (**C**) Tuning width, $w_i$, as a function of the neural density, $d(c_i)$, for three values of $\bar{R}$; Pearson correlation coefficient $\rho$ = (−0.60, −0.87, −0.87). (**D**) MSE (MAP estimate) as a function of $x$ (dashed curves), and power-law fits (solid curves, $R^2$ = (0.90, 0.91, 0.97), $\gamma_e$ = (2.09, 2.10, 1.89)), for three values of $\bar{R}$. (**E**),(**F**) MSE as a function of the neural density (E) and tuning width (F), for three values of $\bar{R}$; Pearson correlation coefficient $\rho_{density}$ = (−0.86, −0.94, −0.86), $\rho_{width}$ = (0.39, 0.29, 0.66).
(PDF)

**S9 Fig. Population coding model with bell-shaped tuning curves.** (**A**) A one-dimensional stimulus is encoded through bell-shaped tuning curves. The number of neurons whose preferred positions are a given stimulus, $x_i$, is denoted by $n_i$, while $w_i$ denotes the tuning width. (**B**) Approximate scaling of the error in stimulus estimate, $\Delta x_i$, when the response of a neuron, with mean $f_j$, is affected by a noise of standard deviation $\eta$.
(PDF)

**S1 Appendix. Properties of the solutions to the minimax problem and maximization of the ELBO.** Demonstrations of the 3 properties of the solutions to the minimax problem of the Lagrangian, Eq (22), we listed in Materials and methods.
(PDF)

**S2 Appendix. Numerical approaches in the case of large neural populations.**
(PDF)

**S3 Appendix. Optimally heterogeneous allocation of neural resources.** We provide an alternative derivation, based on scaling arguments, of the results in Ref. [10].
(PDF)

## Acknowledgments

We thank Luigi Gresele and Giancarlo Fissore for useful discussions, and Trang-Anh Nghiem and Luc Stebens for comments on a earlier version of the manuscript.

## Author Contributions

**Conceptualization:** Simone Blanco Malerba, Michael Woodford, Rava Azeredo da Silveira.

**Formal analysis:** Simone Blanco Malerba, Aurora Micheli, Rava Azeredo da Silveira.

**Funding acquisition:** Michael Woodford, Rava Azeredo da Silveira.

**Investigation:** Simone Blanco Malerba, Aurora Micheli.

**Methodology:** Simone Blanco Malerba, Rava Azeredo da Silveira.

**Project administration:** Rava Azeredo da Silveira.

**Resources:** Rava Azeredo da Silveira.

**Software:** Simone Blanco Malerba, Aurora Micheli.

**Supervision:** Michael Woodford, Rava Azeredo da Silveira.

**Validation:** Simone Blanco Malerba.

**Visualization:** Simone Blanco Malerba.

**Writing – original draft:** Simone Blanco Malerba, Aurora Micheli.

**Writing – review & editing:** Simone Blanco Malerba, Michael Woodford, Rava Azeredo da Silveira.

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
