## [Decision Letter · Decision Letter 0]

11 Oct 2023

Dear Dr. Blanco Malerba,

Thank you very much for submitting your manuscript "Jointly efficient encoding and decoding in neural populations" for consideration at PLOS Computational Biology.

As with all papers reviewed by the journal, your manuscript was reviewed by members of the editorial board and by four independent reviewers. In light of the reviews (below this email), we would like to invite the resubmission of a significantly-revised version that takes into account the reviewers' comments.

All reviewers find your work novel, well-written, and of general interest to the computational neuroscience community. However, we agree with reviewers R2, R3, and R4 that the form and purpose of the generative model in the proposed coding framework requires a more thorough explanation and clarification, as it is somewhat unconventional and thus potentially confusing. In that context, R1 also raises the important question of what limitations the chosen Gaussian model imposes on generalizability. Furthermore, we echo the concerns (mainly by R2 and R3) that the manuscript would benefit from a more thorough discussion of the conceptual advances over previous related work, and its broader impact on our understanding of neural representations of sensory information in the brain. Finally, it is important that a revision resolves the concerns of R3 regarding a potentially  incorrect model-to-data comparison.

Please, also carefully consider all other specific questions, comments, and suggestions in your revision of the manuscript. We cannot make any decision about publication until we have seen the revised manuscript and your point-by-point response to the reviewers' comments. Your revised manuscript will likely be sent back to the reviewers for further evaluation.

Sincerely,

Alan Alfred Stocker, Ph.D.

Guest Editor

PLOS Computational Biology

Lyle Graham

Section Editor

PLOS Computational Biology

Reviewer's Responses to Questions

**Comments to the Authors:**

Reviewer #1: The review is uploaded as an attachment.

Reviewer #2: Previous theoretical studies of efficient coding have been based on optimizing tuning functions for a given stimulus distribution under the assumption that decoding is optimal, which allows the objective function to be specified in terms of Fisher Information. The present study takes a different approach by optimizing tuning functions of the encoder simultaneously with the network parameters of a decoder, which has the architecture of a variational auto-encoder.

This is a polished manuscript describing a substantial and thorough evaluation of the proposed model. However, there is room to make clearer what contribution the findings make to the field. One key conclusion seems to be that, when the rate (R) is sufficiently high – meaning that the decoder can make use of stimulus-evoked variation in neural activity – the optimal tuning comes to resemble that obtained in previous studies (e.g. Ganguli & Simoncelli) that assumed an optimal decoder. For lower R, the solutions become degenerate and the ability for the decoder to reconstruct the stimulus declines. The former state seems to better correspond to neurophysiology, so what conclusions about the brain can be drawn from the latter?

Two significant simplifications in the modelling could benefit from further comment. The impact of basing the model on only a very small number of neurons is investigated to a limited extent, but it would be helpful for the authors to make predictions for scaling up the model based on that investigation – in particular, I wonder if the degeneracy of solutions observed at low R would dissipate also as N was increased? The other assumption is that each unit spikes once or not at all – arguably a neurophysiological "decoder" would integrate spikes over some longer time period: do the authors think this simplification could have influenced their results?

My remaining comments suggest ways in which the clarity of presentation could be improved:

Fig 1: the legend refers to rate and distortion terms, but there are no corresponding labels in the figure. It is not clear what is referred to as "a Boltzmann machine" in the legend (the decoder?) - either clarify or remove this reference.

"The generative model maps neural activity...to parameters of a Gaussian distribution over stimuli" - I understand this is a typical use of the term in ML but in the cognitive sciences "generative model" usually refers to an internal model of how latent states map to stimuli. For the understanding of readers who expect the latter meaning, please address this potential confusion directly at an early point in the manuscript.

Do color patches in Figs 4B,D,E reflect variability across different initializations of parameters, e.g. SEs? Do optimal tuning functions/parameters illustrated in Figs 4C, 8B etc reflect a single initialization or are they averaged in some way? It is hard to assess to what extent the variation with R in these plots reflects real changes in the optimum versus random differences in initialization of simulated parameters. Similarly in 8C it is difficult to interpret the fits without an indication of the uncertainty due to simulation noise.

For the numerical simulations, the initialization of encoder parameters \\theta is described, but not the decoder parameters \\phi. Please explain.

Reviewer #3: The authors propose a novel approach to efficient coding, expanding the usual scheme to include a separate encoding and decoding structure. Here they use a simple tuning-curve-based encoder and a flexible decoder, in contrast with the more common approach to use a complex encoder and a simple decoder (or a simple encoder and simple decoder). The paper describes degeneracies in the results, and explores how the family of solutions are parameterized by a rate-distortion tradeoff. They also compare performance to psychophysical sensitivity in an auditory discrimination task.

Overall I find this to be a nice paper, providing a new perspective on a classic problem, and deriving interesting behaviors from appropriate principles. It sets up an effective combination of simplicity of analysis and modest complexity from a shallow two-layer neural network. The resultant optimized tuning curves have some surprising behaviors. I especially like their principled derivation of the ß-VAE.

Concerns:

For me the most important missing ingredient is any addressing of generalization. Generalization is THE critical property that motivates generative models, and models with different inductive biases (such as they uncover as a function of the rate-distortion tradeoff) should generalize differently.

Concretely, in the scalar case, the authors seem only to evaluate performance on sampled training data. It seems that the values of their otherwise degenerate solutions would be different if they evaluated performance on resampled testing data. The smoother solution of Figure 3, top, may well generalize better to new data than the wiggly solution of Figure 3, bottom. The authors claim that these are equally accurate (L312), but that seems implausible to me based on their curves. I suspect that generalization of these sorts is a pretty minor effect when we’re talking about scalar stimuli, so it may be that they need to move to a more complex framework to demonstrate bigger impacts of their tradeoffs for generalization. Nonetheless, since this is such a fundamental issue it should be discussed.

I think that their explanation of their system could be improved. In particular, I had to read through the paper multiple times before I understood the logic of their approach, and even though I am quite familiar with generative models in neuroscience. Relatedly, I found that their Figure 1 was not as helpful as it could be. Perhaps one option would be to provide more text within the figure to articulate the goals of the optimization, and the nature of the incompatible approximations. One thing that was unfamiliar to me was that their encoder is so simple, using only tuning curves. It seems quite different from such models as the Helmholtz Machine (HM), where the discriminator has comparable sophistication as the generative model, so it makes sense for them to train each other. Additionally, their approach only aims to match the overall stimulus distributions, rather than the latent variables themselves. It seems that these issues could be explained more thoroughly and pedagogically. Relating their approach to more approaches by others (such as the HM or other generative models) would be useful.

I find the application to auditory data unconvincing. There are not enough controls or comparisons to alternative hypotheses to learn much about whether humans are consistent or inconsistent with the authors’ theory. For example, their results are robust to various max allowed rates, which makes me wonder if these results not also compatible with conventional efficient coding?

Additionally, I’m not sure that the authors have correctly compared their theory to data. In Figure 8C the authors compare limens to performance estimated from their model. However, I am concerned that they may have made a unit error that would substantially affect their match. In particular, just noticeable differences between stimuli are measured in units of the stimulus — here, frequency. Yet they compare this directly to Mean Squared Error, which has units of stimulus^2. Shouldn’t they use RMSE (root mean squared error)? If so, this would change the slope in Figure 8C by a factor of 1/2, which would destroy the agreement between their model and data.

Minor comments:

Figure 1: I find it confusing to use \\hat{x} in a posterior. The posterior is really over x, not over estimates \\hat{x} (unless \\hat{x} defines a new possibly wrong latent variable, which I don’t think they do — and even in that case I wouldn’t use \\hat{x} because it has connotations of an estimate). An estimate \\hat{x} would be a function of the posterior over x.

Similarly, their figure uses D(x,\\hat{x}) in the figure, but not in the text.

How does the distortion depend on the uncertainty in q, as opposed to a point estimate xh?

Why are they computing errors from SAMPLES from q(x|r), instead of errors of an estimate based on q(x|r), like the ML or MAP? Those errors will have terms arising from the sampling, which would be more error than needed based upon q.

How can their wiggly generative distribution in Figure 3, bottom, have 17 peaks when there are only 10 tuning functions that are being combined?

Why does the distortion in Figure 4B decrease and then increase? I’d expect that with higher rates the distortion should monotonically decrease?

L524: “In previous studies, internal models were defined by conditioning the probability of a stimulus, x, on the realization of a latent variable, z, through their joint distribution, q(z, x) = q(x|z)q(z). How the latent variable was related to a specific neural representation was not prescribed.”

First, in these sampling models, z must be linear in r, which is actually a very strong constraint.

Second, work from Ralf Haefner’s group has done more along these lines, using an explicit model for r akin to the classic sparse coding approach of Olshausen and Field. Additionally, multiple works by Jeff Beck and colleagues has used generative models to derive specific neural representations.

It would be nicer for storytelling if their paper ended on note that discusses something like future prospects of their model, instead of trailing off with a critique of other suboptimal models.

Reviewer #4: This paper presents a new theoretical framework that synthesizes the efficient coding hypothesis, which argues that the nervous system seeks to maximize information about the environment, and generative modeling approaches, which seek to frame neural coding as inferring the hidden causes underlying sensory inputs under a generative model of the world. This novel approach takes the form of a variational auto-encoder, a generative modeling approach developed originally in the machine learning literature, which contains both an encoder (for approximate inference under an intractable generative model) and a decoder (the generative model itself). The work is novel and interesting, and likely to be of broad interest to the theoretical and computational neuroscience community. I enjoyed reading it, and think it is a highly appropriate for the readership of PLoS CB. Most of my comments focus on technical aspects of the work and issues of clarity and completeness, which I will outline in more detail below. Overall it is a very nice paper, and I congratulate the authors on this impressive contribution to the literature.

Detailed Comments:

--- line 6: " This hypothesis has been successful in predicting neural responses to natural stimuli in various sensory areas [2–4]." I would also cite Laughlin's seminal 1981 paper here (which as far as I'm aware is the first paper to propose an empirical test of the efficient coding hypothesis):

Laughlin, S. B. "A simple coding procedure enhances a neuron's information capacity" Z. Naturforsch, 1981, 36, 910-912.

--- line 28: "As opposed to the efficient coding approach, which prescribes a stochastic mapping from stimulus to neural activity, the generative model approach prescribes a stochastic mapping from neural activity to stimulus. This mapping implies a posterior distribution on neural activity, which can be read off from neural data."

This is a complex idea that I think may be unfamiliar / confusing to some readers -- I might suggest unpacking this a bit more. (i.e., who does the generative modeling approach prescribe a stochastic mapping from neural activity to stimulus?)

--- line 40: "structure of a variational autoencoder (VAE) [17]."

I believe VAEs were proposed independently in two papers that appeared right around the same time, and so it might be nice to cite both of them. In addition to ref 17 (Kingma & Welling 2014), you might add:

Rezende, D. J., S. Mohamed, and D. Wierstra (2014). “Stochastic backpropagation and approximate

---

## [Decision Letter · Decision Letter 1]

7 Jun 2024

Dear Dr. Blanco Malerba,

We are pleased to inform you that your manuscript 'Jointly efficient encoding and decoding in neural populations' has been provisionally accepted for publication in PLOS Computational Biology.

Best regards,

Alan Alfred Stocker, Ph.D.

Guest Editor

PLOS Computational Biology

Lyle Graham

Section Editor

PLOS Computational Biology

Reviewer's Responses to Questions

**Comments to the Authors:**

Reviewer #1: Thank the authors for the thorough revision of the manuscript. All my previous comments have been addressed. This paper will be of great interest to the neural coding community, and I am looking forward to future results of generalizing this approach to more complex encoding models and sensory stimuli.

Reviewer #2: My thanks to the authors for their very thorough response to my review - I have no further comments.

Reviewer #4: The authors have done a nice job addressing my comments, and I thank them for their thorough and detailed response. I believe the paper is substantially improved and I have no further comments.

**Have the authors made all data and (if applicable) computational code underlying the findings in their manuscript fully available?**

Reviewer #1: Yes

Reviewer #2: Yes

Reviewer #4: Yes

PLOS authors have the option to publish the peer review history of their article (what does this mean?). If published, this will include your full peer review and any attached files.

Reviewer #1: No

Reviewer #2: No

Reviewer #4: No

---

## [Editor Report · Acceptance letter]

3 Jul 2024

PCOMPBIOL-D-23-00970R1 

Jointly efficient encoding and decoding in neural populations

Dear Dr Blanco Malerba,

I am pleased to inform you that your manuscript has been formally accepted for publication in PLOS Computational Biology. Your manuscript is now with our production department and you will be notified of the publication date in due course.

With kind regards,

Olena Szabo
